# LLM Unlearning Under the Microscope:
# A Full-Stack View on Methods and Metrics

## Abstract

Machine unlearning for large language models (LLMs) aims to remove *undesired* data, knowledge, and behaviors (*e.g.*, for safety, privacy, or copyright) while preserving useful model capabilities. Despite rapid progress over the past two years, research in LLM unlearning remains fragmented, with limited clarity on what constitutes effective unlearning and how it should be rigorously evaluated. In this work, we present a principled taxonomy of *twelve* recent stateful unlearning methods, grouped into three methodological families: *divergence-driven optimization*, *representation misalignment*, and *rejection-based targeted unlearning*. Building on this taxonomy, we revisit the evaluation of unlearning effectiveness (UE), utility retention (UT), and robustness (Rob), focusing on the WMDP benchmark. Our analysis shows that current evaluations, dominated by multiple-choice question (MCQ) accuracy, offer only a narrow perspective, often overstating success while overlooking the model's actual generation behavior. To address this gap, we introduce open question-answering (Open-QA) metrics that better capture generative performance and reveal the inherent UE–UT tradeoff across method families. Furthermore, we demonstrate that robustness requires finer-grained analysis: For example, vulnerabilities differ substantially between *in-domain relearning* and *out-of-domain fine-tuning*, even though both fall under model-level attacks. Through this study, we hope to deliver a full-stack revisit of LLM unlearning and actionable guidance for designing and evaluating future methods.

## 1 Introduction

With the rapid advances of large language models (LLMs), their tendency to memorize and regurgitate training data has raised serious concerns about privacy, safety, and intellectual property (Liu et al., 2024d; Che et al., 2025; Barez et al., 2025). More broadly, from the perspective of generative behavior, sensitive information, harmful content, and copyrighted material can be unintentionally reproduced, motivating the urgent need for *machine unlearning*. The goal of unlearning is to selectively remove undesired data, knowledge, or behaviors from a trained model while preserving its general utility for normal tasks (Yao et al., 2024; Maini et al., 2024; Liu et al., 2024b; Wang et al., 2025a).

The growing interest in LLM unlearning has spurred the development of a wide range of algorithms. In this work, we examine **twelve** representative methods and categorize them into three families: divergence-driven optimization, representation misalignment, and rejection-based targeted unlearning. Divergence-driven optimization methods drive the model away from a reference distribution (Yao et al., 2024; Fan et al., 2025; Wang et al., 2025b), with negative preference optimization (NPO) (Zhang et al., 2024a) and simple NPO (SimNPO) (Fan et al., 2024a) as examples. Representation-misalignment methods disrupt the embeddings of forget data relative to their reference model representations (Zou et al., 2024; Tamirisa et al., 2024; Sheshadri et al., 2024), such as representation misdirection for unlearning (RMU) (Li et al., 2024). Rejection-based methods enforce unlearning by producing explicit rejection responses to forget queries (Yuan et al., 2024; Singh et al., 2025), with I Don't Know (IDK) (Maini et al., 2024) as a representative case.

These diverse LLM unlearning approaches are typically evaluated on off-the-shelf benchmarks. A common choice is the Weapons of Mass Destruction Proxy (WMDP) benchmark (Li et al., 2024), valued for its practical focus on erasing harmful generation behaviors and for not requiring prior fine-tuning on the forget set. In WMDP, unlearning performance is assessed along two dimensions:

*unlearning effectiveness* (UE) and *utility retention* (UT). However, these evaluations primarily rely on multiple-choice questions (MCQ), where the model selects from predefined options. The risk is that such evaluations may *obscure* the actual free-form generation behavior of LLMs post-unlearning, limiting the assessment of UE and UT on forget-relevant and forget-irrelevant queries beyond predefined answer options. In addition, even under the MCQ evaluation metric, insights and comparative analyses across different unlearning method families remain limited. Beyond UE and UT, unlearning robustness (Rob) is also critical, as forgotten knowledge can re-emerge once the unlearned model is "attacked" (Hu et al., 2024; Łucki et al., 2024; Tamirisa et al., 2024; Che et al., 2025; Lynch et al., 2024). Attacks take various forms, ranging from model-level (Hu et al., 2024; Tamirisa et al., 2024; Fan et al., 2025; Wang et al., 2025b; Zhang et al., 2024b) to input-level (Łucki et al., 2024; Lynch et al., 2024). However, systematic studies of unlearning robustness and its relationship to these different attacks also remain lacking.

Given the diversity of unlearning methods and the incompleteness of current evaluations, the key research question we aim to address is:

> *(Q) Can we conduct a full-stack investigation of LLM unlearning that yields methodology-wise insights across all key metrics (UE, UT, and Rob)?*

To tackle (Q), we first draw methodological insights from our proposed categorization: *divergence-driven optimization*, *representation misalignment*, and *rejection-based targeted unlearning*. We then revisit the conventional unlearning assessments, UE and UT, and argue that they should also be evaluated through open question answering (Open-QA), where the model generates free-form responses, beyond MCQ. Relying solely on MCQ can lead to a myopic view of UE and UT across different unlearning methods, while Open-QA provides an important complementary perspective. For instance, MCQ-based MMLU assessments of UT *cannot* fully capture the over-forgetting issue in divergence-driven optimization methods such as NPO (Zhang et al., 2024a), nor can MCQ-based UE evaluations fully reflect the performance of rejection-based targeted unlearning. Furthermore, in the dimension of Rob, we find that robustness in LLM unlearning should be examined at a finer granularity, including (i) in-domain relearning (Hu et al., 2024; Fan et al., 2025), where the model is fine-tuned on a subset of forget data, and (ii) out-of-domain fine-tuning (Wang et al., 2025b), where the model is adapted to unrelated downstream tasks. Divergence-driven optimization methods are generally more resilient to in-domain relearning, whereas representation misalignment methods show stronger resistance to out-of-domain fine-tuning. In addition, robustness against input-level attacks such as jailbreaking is more closely aligned with in-domain relearning.

Prior work has made initial attempts to study LLM unlearning *systematically*, focusing on robustness (Che et al., 2025; Hu et al., 2025) and evaluation (Feng et al., 2025). Our work advances these efforts with three key novelties: (i) a methodological categorization that guides a rethinking of LLM unlearning, (ii) evaluation of UE and UT across both answer selection and free-form generation, highlighting their interaction with different methods, and (iii) an analysis of unlearning robustness that examines diverse forms of model-level weight perturbations and their connections to input-level jailbreaking attacks. We summarize **our contributions** below.

• We establish a principled taxonomy of 12 recent unlearning methods, categorizing them into divergence-driven optimization, representation misalignment and rejection-based targeted unlearning.

• We revisit evaluation practices by moving beyond MCQ to incorporate Open-QA metrics, which better capture generative performance and reveal fundamental characteristics of different unlearning method families across UT and UE.

• We revisit the robustness of LLM unlearning by analyzing vulnerabilities under model-level attacks (in-domain relearning, out-of-domain fine-tuning, and quantization) and input-level jailbreak attacks, demonstrating how these robustness dimensions are connected.

## 2 RELATED WORK

**LLM unlearning and benchmarking studies.** Recent work on LLM unlearning (Liu et al., 2024b; Maini et al., 2024; Liu et al., 2024a; Yao et al., 2023) has made progress in mitigating risks such as copyright infringement (Eldan & Russinovich, 2023), privacy leakage (Hu et al., 2024; Wu et al.,

2023), and harmful content generation (Li et al., 2024; Lu et al., 2022). For detailed introductions to the diverse families of unlearning methods, we refer readers to Sec. 3.

Given the growing interest in LLM unlearning, several benchmarks have been proposed to systematically evaluate its effectiveness (Li et al., 2024; Maini et al., 2024; Jin et al., 2024; Shi et al., 2024; Eldan & Russinovich, 2023). These benchmarks can be grouped into two categories based on whether models are fine-tuned on the forget corpus. The first category fine-tunes models on domain-specific corpora to introduce unlearning targets. WHP (Eldan & Russinovich, 2023) uses the Harry Potter series, TOFU (Maini et al., 2024) constructs synthetic author profiles, and MUSE (Shi et al., 2024) provides Harry Potter and news corpora with privacy-leakage evaluation. A drawback is that domain-specific fine-tuning can degrade general abilities such as reasoning, complicating fair utility assessment (Jin et al., 2024). The second category avoids fine-tuning, aligning more closely with real-world use cases. WMDP (Li et al., 2024) constructs forget corpora in biology, chemistry, and cybersecurity, and evaluates unlearning via multiple-choice QA proxies for hazardous knowledge, alongside utility on MMLU (Hendrycks et al., 2020) and MT-Bench (Zheng et al., 2023). RWKU (Jin et al., 2024) focuses on real-world entities, evaluating memorization on the forget set and reasoning, truthfulness, factuality, and fluency on retain tests.

**Adversarial robustness of LLM unlearning.** Robustness has emerged as a central challenge for unlearning, as unlearned models remain vulnerable to diverse attacks (Łucki et al., 2024; Che et al., 2025; Hu et al., 2024). Parameter-level attacks can restore forgotten content through light fine-tuning (Łucki et al., 2024; Hu et al., 2024; Qi et al., 2023; Halawi et al., 2024; Lermen et al., 2023; Huang et al., 2024), pruning (Jain et al., 2023; Wei et al., 2024; Lee et al., 2018), or quantization (Zhang et al., 2024b), which may re-expose knowledge by compressing weights. Representation-level attacks perturb embeddings or hidden activations to revive residual traces (Schwinn et al., 2024; Sheshadri et al., 2024), while input-level attacks exploit adversarial prompting or query optimization (Chao et al., 2025; Shin et al., 2020), ranging from gradient-guided search (Zou et al., 2023) to perplexity-based strategies (Sadasivan et al., 2024). In response, several defenses have been proposed. Adversarial training strengthens robustness against such attacks (Sheshadri et al., 2024), and meta-learning strategies further enhance defense (Tamirisa et al., 2024; Sondej et al., 2025). Sharpness-aware minimization encourages flat minima to reduce susceptibility to relearning (Fan et al., 2025). Invariance-based regularization introduces robustness through invariant risk principles (Wang et al., 2025b), while distillation-based methods transfer knowledge into partially noised student models (Lee et al., 2025). Although robustness benchmarks are emerging (Che et al., 2025; Hu et al., 2025), they remain limited: Che et al. (2025) evaluates only with MCQ accuracy, and Hu et al. (2025) considers only in-domain relearning, leaving comprehensive assessment open.

## 3 A Taxonomy of Stateful Unlearning Methods: Methodologies, Design Principles, and Insights

**Problem setup for LLM unlearning.** The unlearning problem is defined with respect to a subset of data instances that must be erased, denoted as the *forget set* ($\mathcal{D}_\mathrm{f}$). To preserve model utility, one also specifies a complementary *retain set* ($\mathcal{D}_\mathrm{r}$) whose knowledge should remain unaffected. These two sets capture the dual objectives of unlearning: removing information from $\mathcal{D}_\mathrm{f}$ while maintaining the model's general utility as reflected by $\mathcal{D}_\mathrm{r}$. This trade-off can be formulated as a regularized optimization problem (Liu et al., 2024b):

$$\underset{\boldsymbol{\theta}}{\text{minimize}} \quad \ell_\mathrm{f}(\boldsymbol{\theta}; \mathcal{D}_\mathrm{f}) + \lambda \ell_\mathrm{r}(\boldsymbol{\theta}; \mathcal{D}_\mathrm{r}), \tag{1}$$

where $\boldsymbol{\theta}$ are the model parameters to be updated, $\lambda \geq 0$ is a regularization weight balancing forgetting and retention, and $\ell_\mathrm{f}$ and $\ell_\mathrm{r}$ denote loss terms for the forget and retain objectives, respectively.

A large body of work has investigated different design choices for instantiating (1) (Yao et al., 2023; Zhang et al., 2024a; Fan et al., 2024a; 2025; Wang et al., 2025b; Li et al., 2024; Zou et al., 2024; Gandikota et al., 2024; Sheshadri et al., 2024; Tamirisa et al., 2024; Yuan et al., 2024; Singh et al., 2025). Based on their methodological principles, we categorize existing approaches into three families: *divergence-driven optimization*, *representation misalignment*, and *rejection-based targeted unlearning*. See detailed insights below.

**Divergence-driven optimization for unlearning.** The first family of methods designs the forget loss $\ell_\mathrm{f}$ in (1) to maximize the "divergence" between the prediction logits of the unlearned model

and the reference (original) model on the forget set $\mathcal{D}_\text{f}$. We refer to this class as *divergence-driven optimization*, since it explicitly drives the model away from the reference model.

The most basic instance is gradient ascent (GA) (Thudi et al., 2022), which directly increases the prediction loss on $\mathcal{D}_\text{f}$. However, GA often pushes the model too far from the reference, leading to collapse (Zhang et al., 2024a). To address this, several GA-type variants have been proposed. Gradient difference (**GradDiff**) (Yao et al., 2023) balances objectives by applying gradient ascent on $\mathcal{D}_\text{f}$ while using gradient descent on $\mathcal{D}_\text{r}$, thereby controlling divergence from the reference. Negative preference optimization (**NPO**) (Zhang et al., 2024a) interprets forget samples as negative preferences within the DPO (direct preference optimization) (Rafailov et al., 2024). This specifies $\ell_\text{f}$ with

$$\ell_\text{NPO}(\boldsymbol{\theta}) = \mathbb{E}_{(x,y)\in\mathcal{D}_\text{f}} \left[ -\tfrac{2}{\beta} \log \sigma \left( -\beta \log \tfrac{\pi_{\boldsymbol{\theta}}(y|x)}{\pi_\text{ref}(y|x)} \right) \right], \tag{2}$$

where $\pi_{\boldsymbol{\theta}}(y \mid x)$ represents the prediction probability of the model $\boldsymbol{\theta}$ given the input-response pair $(x, y)$, and $\pi_\text{ref}$ refers to the reference model. Simple NPO (**SimNPO**) (Fan et al., 2024a) further mitigates reference-model bias in NPO by modifying $\ell_\text{f}$ to $-\tfrac{2}{\beta} \log \sigma \left( -\tfrac{\beta}{|y|} \log \pi_{\boldsymbol{\theta}}(y \mid x) \right)$.

Building on NPO, recent works have further addressed robustness gaps in LLM unlearning, particularly sensitivity to model-level attack after unlearning. Examples include **NPO+SAM** (Fan et al., 2025), which leverages sharpness-aware minimization (SAM) (Foret et al., 2021) to flatten the forget loss landscape, and **NPO+IRM** (Wang et al., 2025b), which applies invariant risk minimization (IRM) (Arjovsky et al., 2019) to encourage robustness across distributional variations.

**Representation misalignment for unlearning.** Another class of methods operates on internal representations, seeking to misalign the representations of $\mathcal{D}_\text{f}$ with their original representations in the reference model. We term this family *representation misalignment*. The principle originates from early methods (Golatkar et al., 2020; Fan et al., 2024b) that inject randomness into forget data to disrupt memorization and reduce alignment on $\mathcal{D}_\text{f}$. The most popular approach wihtin this family is representation misdirection for unlearning (**RMU**) (Li et al., 2024), where the hidden states of the unlearned model are mapped to a random vector. This formulates the forget loss $\ell_\text{f}$ as

$$\ell_\text{RMU}(\boldsymbol{\theta}) = \mathbb{E}_{x\in\mathcal{D}_\text{f}} \left[ ||M_{\boldsymbol{\theta}}(x) - c \cdot \mathbf{u}||_2^2 \right], \tag{3}$$

where $M_{\boldsymbol{\theta}}$ represents certain intermediate-layer representations of $\boldsymbol{\theta}$, $c > 0$ is a hyperparameter that controls activation scaling, and $\mathbf{u}$ is a random vector drawn from a standard uniform distribution.

In addition, representation rerouting (**RR**) (Zou et al., 2024) modifies RMU by replacing the $\ell_2$ norm in $\ell_f$ with cosine similarity between the unlearned model $M_{\boldsymbol{\theta}}$ and the reference model. Other approaches, such as tampering attack resistance (**TAR**) (Tamirisa et al., 2024) and latent adversarial training (**LAT**) (Sheshadri et al., 2024), build on RMU with meta-learning or adversarial training in the latent space to improve unlearning robustness.

**Rejection-based unlearning.** Unlike divergence-driven or representation-misalignment approaches, which perform *untargeted* unlearning by discouraging alignment with the forget set, the *rejection-based targeted unlearning* family enforces *targeted* unlearning through explicit rejection responses to forget queries. A representative method is the I Don't Know (**IDK**) strategy (Maini et al., 2024), which defines the forget loss $\ell_\text{f}$ as the prediction loss over rejection-labeled forget data:

$$\ell_\text{IDK}(\boldsymbol{\theta}) = \mathbb{E}_{(x\in\mathcal{D}_\text{f}, y\in\mathcal{D}_\text{IDK})} \left[ -\log \pi_{\boldsymbol{\theta}}(y \mid x) \right], \tag{4}$$

where $\mathcal{D}_\text{IDK}$ denotes the set of rejection labels expressed in different formats.

Beyond the simplest IDK formulation, **DPO** (Rafailov et al., 2024; Zhang et al., 2024a) can also be adapted for unlearning by treating rejection as the positive response for forget data. Other extensions include **IDK+AP** (Yuan et al., 2024), which, similar in spirit to DPO, introduces an answer preservation (AP) loss that regards the normal response as positive on retain data and the rejection response as positive on forget data, thereby augmenting IDK with an additional alignment objective. Similarly, erasing via language modeling (**ELM**) (Gandikota et al., 2024) aligns the unlearned model's outputs with those of a prompted reference model, using a predefined prefix (*e.g.*, *"As a novice in bioweapons"*) to steer responses toward refusal-like outputs.

**Benchmarks and evaluations.** The aforementioned methods have been validated under different unlearning benchmarks, such as WMDP for hazardous knowledge removal (Li et al., 2024), MUSE for copyrighted content removal (Shi et al., 2024), TOFU for fictional data removal (Maini et al., 2024),

WHP for "Harry Potter" book series knowledge (Eldan & Russinovich, 2023), PKU-SafeRLHF for harmful content removal (Ji et al., 2024) and Circuit Breaker for toxic content removal (Zou et al., 2024).

Although no consensus exists on the *most appropriate* benchmarks for unlearning, we adopt **WMDP** (Li et al., 2024) for its focus on erasing harmful knowledge without requiring extra fine-tuning on the $\mathcal{D}_f$. Given the common use of WMDP-Bio (which contains biological knowledge for harm reduction), we refer to WMDP as WMDP-Bio. For supplementary validation, we may also consider **MUSE** (Shi et al., 2024).

Unlearning performance is most commonly evaluated in terms of unlearning effectiveness (**UE**) and utility retention (**UT**). The UE and UT evaluation metrics in existing benchmarks can be generally classified into two types: *(i) multiple-choice questions (MCQ)*, where the model selects from predefined options, and *(ii) open question-answering (Open-QA)*, where the model generates free-form answers. Therefore, we denote $UE_{MCQ}$ (or $UT_{MCQ}$) and $UE_{Open-QA}$ (or $UT_{Open-QA}$) as the

Table 1: Taxonomy of 12 unlearning methods grouped into *divergence-driven optimization*, *representation misalignment*, and *rejection-based targeted unlearning* families. A checkmark (✓) marks incorporation of *robust unlearning*. Methods are evaluated on benchmarks WMDP (W), MUSE (M), TOFU (T), WHP (H), PKU-SafeRLHF (P) and Circuit Breaker (C), using unlearning effectiveness (UE), utility retention (UT), and robustness (Rob). UE and UT are measured by multiple-choice (MCQ) or open question-answering (Open-QA), while Rob is tested at model-level or input-level. Benchmark abbreviations denote both methods and evaluation settings.

| Method | Reference | Robust Design | UE | | UT | | Rob | |
|---|---|---|---|---|---|---|---|---|
| | | | MCQ | Open-QA | MCQ | Open-QA | Model | Input |
| *Divergence-driven optimization* | | | | | | | | |
| **GradDiff** | (Yao et al., 2023) | ✗ | | P | | P | | |
| NPO | (Zhang et al., 2024a) | ✗ | | T | | T | | |
| **SimNPO** | (Fan et al., 2024a) | ✗ | W | T/M | W | T/M | T | |
| NPO+SAM | (Fan et al., 2025) | ✓ | W | M | W | M | W/M | W |
| NPO+IRM | (Wang et al., 2025b) | ✓ | W | M | W | M | W/M | |
| *Representation misalignment* | | | | | | | | |
| **RMU** | (Li et al., 2024) | ✗ | W | | W | W | | W |
| RR | (Zou et al., 2024) | ✗ | | C | C | C | | C |
| RMU+LAT | (Sheshadri et al., 2024) | ✓ | | W/H | W/H | W | W | H |
| TAR | (Tamirisa et al., 2024) | ✓ | W | | W | | W | |
| *Rejection-based targeted unlearning* | | | | | | | | |
| ELM | (Gandikota et al., 2024) | ✗ | W/H | | W/H | W/H | | W |
| DPO | (Zhang et al., 2024a) | ✗ | | T | T | T | | |
| IDK+AP | (Yuan et al., 2024) | ✗ | | T | T | T | | |

corresponding UE (or UT) metrics, respectively. In addition to UE and UT, robustness (**Rob**) has emerged as another critical dimension in evaluating LLM unlearning. We distinguish two types of robustness assessments: *model-level attacks*, such as in-domain relearning (Hu et al., 2024; Fan et al., 2025) and out-of domain fine-tuning (Wang et al., 2025b); and *input-level attacks*, such as jailbreaking attack (Łucki et al., 2024).

**Table 1** summarizes 12 unlearning approaches with their benchmark applications and evaluation metrics, where benchmark abbreviations denote both methods and datasets. **Appendix A** outlines key implementation details. In the remainder, we revisit these approaches to assess unlearning evaluation (Sec. 4) and robustness (Sec. 5), uncovering overlooked insights into LLM unlearning.

## 4 BEYOND ANSWER SELECTION: RETHINKING UNLEARNING EVALUATION FOR EFFECTIVENESS AND UTILITY

> **Summary of insights into unlearning effectiveness and utility retention**
>
> **(1)** Unlearning evaluation should *go beyond answer selection* (*i.e.*, highest-probability choice in WMDP) to assess actual generation content in both UE and UT.
> **(2)** *Divergence-driven optimization* methods often *over-forget*, breaking generation on forget queries. In contrast, *representation misalignment* methods better preserve generation.
> **(3)** For *rejection-based targeted unlearning*, evaluating beyond answer selection is essential to capture UT. Moreover, aggressive fine-tuning on rejection targets can harm UT.
> **(4)** UE-UT *tradeoffs* are fundamental but often hidden by MCQ-based evaluation. Over-forgetting especially degrades utility on Open-QA tasks that rely on generated content.

In this section, we revisit unlearning evaluation across the UE (unlearning effectiveness) and UT (utility retention) dimensions for the methodological families in Table 1. We contrast answer selection-based evaluations (*i.e.*, MCQ tasks) with content-based evaluations (*i.e.*, Open-QA tasks). Recall that

in MCQ settings, the model selects the option with the highest predicted score, whereas Open-QA requires free-form generation. At the start of this section, we summarized the key insights.

**Open-QA as a crucial lens for UE and UT evaluation.** In LLM unlearning on WMDP, UE is typically measured by *accuracy* on the WMDP evaluation set, assessed through MCQ where success is judged by selecting the correct option (*e.g.*, A/B/C/D). Likewise, most unlearning benchmarks (including WMDP) evaluate UT using *MMLU* tasks, which also follow the MCQ format. The main limitation of relying solely on MCQ-based evaluations is that they fail to capture the model's actual generated capabilities after unlearning, leading to a false sense of unlearning success and obscuring the true rationale and quality of unlearning.

In **Table A1** of **Appendix B**, we illustrate the distinction between answer selection and answer generation using NPO/RMU-unlearned Llama-3 8B Instruct on WMDP, evaluated against a forget-relevant query. From the answer selection perspective, the unlearned model (whether NPO or RMU) selects an *incorrect* option (*i.e.*, option $D$, differing from the original model's choice), indicating successful unlearning on WMDP. However, from an answer generation perspective, the model produces *nonsensical text* instead of valid answer choices, revealing that it has internally disrupted its generation ability for forget queries. This also raises concerns of *over-forgetting*, as the degradation may also impair generation on non-forget inputs, which would not be captured by MMLU.

To capture the perspective of answer generation, we propose using Open-QA for evaluating both UE and UT. For UE, we adopt the *entailment score* (**ES**) (Yuan et al., 2024; Yao & Barbosa, 2024; Poliak, 2020), which measures the factual consistency of model outputs against the original pre-unlearned model's response (*i.e.*, the correct answer). A higher ES indicates that the unlearned model can still infer the correct information when queried with forget data. To improve ES compatibility with the unlearned model's output format, we integrate few-shot examples (with the desired answer style) into the forget data prompts to guide generation toward the correct format during evaluation; see **Appendix A** for details. For UT, we recommend incorporating Open-QA tasks, such as **IFEval** (Zhou et al., 2023), **GSM8K** (Cobbe et al., 2021), alongside MCQ tasks including **MMLU** (Hendrycks et al., 2020), **MathQA** (Amini et al., 2019), **TruthfulQA** (Lin et al., 2021). Adding these benchmarks enables a more complete evaluation of utility. In particular, IFEval captures instruct-following ability, and GSM8K measures quantitative reasoning. Together they provide a balanced view of how unlearning affects model utility.

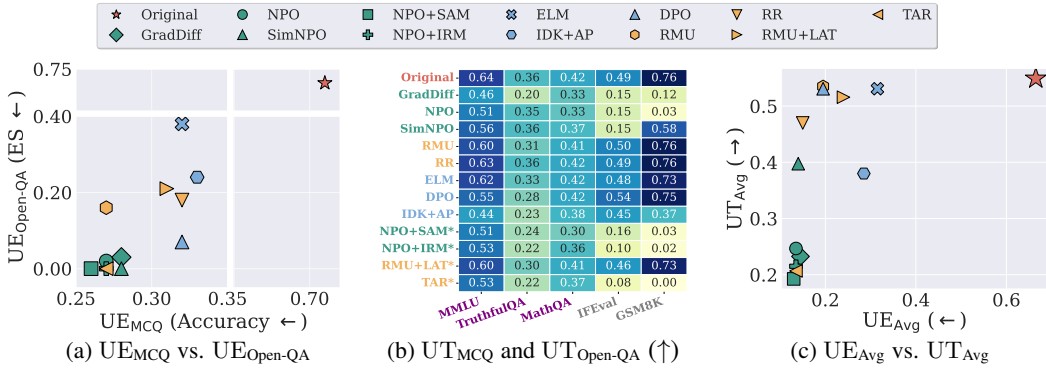

Figure 1: Unlearning effectiveness (UE) and utility retention (UT) evaluation of unlearning methods on WMDP with Llama-3 8B Instruction. (a) $UE_{MCQ}$ denotes accuracy on the WMDP evaluation set, and $UE_{Open-QA}$ denotes ES on the WMDP evaluation set. The arrow direction along each axis indicates the direction of better performance. (b) $UT_{MCQ}$ includes MMLU, TruthfulQA, and MathQA, while $UT_{Open-QA}$ includes IFEval and GSM8K. (c) $UT_{Avg}$ is defined as the mean of $UT_{MCQ}$ and $UT_{Open-QA}$, and $UE_{Avg}$ is defined analogously.

In **Fig. 1-(a)**, we present the $UE_{MCQ}$ and $UE_{Open-QA}$ of 12 unlearning methods along with the original model. As shown, for RMU and NPO, even when the unlearned models achieve the same $UE_{MCQ}$, their $UE_{Open-QA}$ can differ significantly. This highlights the necessity of jointly measuring both $UE_{MCQ}$ and $UE_{Open-QA}$. Moreover, we observe that *divergence-driven optimization* generally achieves better $UE_{MCQ}$ and $UE_{Open-QA}$ compared with other families.

In **Fig. 1-(b)**, we present the $UT_{MCQ}$ and $UT_{Open-QA}$ of 12 unlearning methods along with the original model. As shown, although *divergence-driven optimization* (e.g., NPO) achieves a similar $UT_{MCQ}$ as *representation misalignment* (e.g., RMU), its $UT_{Open-QA}$ is much lower, indicating that NPO over-forgets and thereby reduces its generation capability. Furthermore, compared with RMU, TAR shows

a marked decline in $\mathrm{UT}_{\text{Open-QA}}$, indicating that adding robustness to RMU comes at the expense of utility in Open-QA. In **Fig. A1** of **Appendix B**, analysis of logits shows that NPO achieves unlearning by collapsing logits and inducing over-forgetting, which ultimately impairs generation ability.

In **Fig. 1-(c)**, we report $\mathrm{UE}_{\text{Avg}}$ and $\mathrm{UT}_{\text{Avg}}$. Here, $\mathrm{UT}_{\text{Avg}}$ is computed as the average of $\mathrm{UT}_{\text{MCQ}}$ and $\mathrm{UT}_{\text{Open-QA}}$, and $\mathrm{UE}_{\text{Avg}}$ is defined analogously. A higher $\mathrm{UT}_{\text{Avg}}$ indicates better utility, while a lower $\mathrm{UE}_{\text{Avg}}$ reflects stronger unlearning effectiveness. When considering UE and UT jointly, *representation misalignment* generally outperforms *rejection-based targeted unlearning*, which in turn outperforms *divergence-driven optimization*. Within these families, RMU is the strongest under representation misalignment, DPO under rejection-based targeted unlearning, and SimNPO under divergence-driven optimization.

**Rethinking rejection-based methods.** Compared to divergence-driven optimization and representation misalignment, the *rejection-based targeted unlearning* family is less commonly used in LLM unlearning. Below, we highlight several overlooked insights. *First*, as shown in Fig. 1-(a), rejection-based targeted unlearning exhibit significantly lower $\mathrm{UE}_{\text{MCQ}}$ compared to others. This performance gap is one of the primary reasons for their limited popularity in LLM unlearning (Li et al., 2024; Shi et al., 2024). However, this view may be myopic, as under $\mathrm{UE}_{\text{Open-QA}}$ the performance of rejection-based methods varies much more substantially, with DPO achieving the best $\mathrm{UE}_{\text{Open-QA}}$ among them. *Second*, if we only consider the conventional UT assessment under MMLU in Fig. 1-(b), DPO does not exhibit clearly better MMLU accuracy than divergence-driven optimization methods. Consequently, the prevailing understanding in the literature has been that DPO offers *no advantage* in terms of utility (Maini et al., 2024; Zhang et al., 2024a). However, this view is also myopic, as DPO's $\mathrm{UT}_{\text{Open-QA}}$ metrics (IFEval and GSM8k) remain comparable to those of the original model. *Third*, while IDK+AP and DPO share the objective of eliciting rejection responses (*e.g.*, "I don't know") to forget-relevant queries, IDK+AP shows markedly worse UT, as in Fig. 1-(c). We attribute this degradation to its stricter log-likelihood loss, which overly enforces rejection probabilities. In **Fig. A2** of **Appendix B**, we propose a mitigation for IDK+AP by leveraging the strengths of DPO.

## 5 Multi-Faceted Robustness Assessments for Unlearning

> **Summary of insights into unlearning robustness**
>
> **(1)** From a *model tampering* perspective, robustness against in-domain relearning differs from out-of-domain fine-tuning. For instance, *divergence-driven optimization* methods are typically more resilient to in-domain relearning, while *representation misalignment* methods better withstand out-of-domain fine-tuning.
> **(2)** From a *model quantization* perspective, robustness should be assessed alongside the utility loss caused by compression, to avoid misinterpreting performance gains that simply stem from model incapacity under quantization.
> **(3)** From an *input-level* perspective, *representation misalignment* methods are generally vulnerable to jailbreaking attacks, and robustness to such attacks aligns more closely with in-domain relearning than with out-of-domain fine-tuning.
> **(4)** From an *overall robustness* perspective, including both weight- and input-level perturbations, augmenting unlearning with robust designs, *e.g.*, SAM (Fan et al., 2025), IRM (Wang et al., 2025b), and TAR (Tamirisa et al., 2024), improves resilience, even when these techniques were not explicitly developed to address all unlearning vulnerabilities.

In this section, we investigate the robustness of LLM unlearning methods as categorized in Table 1. Our analysis spans the full spectrum of vulnerabilities, from model-level (*e.g.*, in-domain relearning, out-of-domain fine-tuning, and quantization) to input-level jailbreaking attacks.

**Robustness against in-domain relearning and out-of-domain fine-tuning.** Prior work (Hu et al., 2024; Fan et al., 2025; Wang et al., 2025b; Deeb & Roger, 2024; Hu et al., 2025; Che et al., 2025) has highlighted the vulnerability of unlearned models to model-level attack after unlearning. However, most studies examine only one type: either *in-domain relearning* (on data aligned with the forget set $\mathcal{D}_{\text{f}}$, *e.g.*, subsets of $\mathcal{D}_{\text{f}}$) or *out-of-domain fine-tuning* (adapting to unrelated downstream tasks such as GSM8K for math reasoning). We argue that these correspond to two distinct categories of perturbations, analogous to adversarial robustness versus out-of-distribution robustness. Thus, we propose studying them jointly to obtain a more complete understanding of unlearning robustness.

To this end, we evaluate unlearning performance, measured by answer selection accuracy ($\text{UE}_{\text{MCQ}}$) and ES-based generation assessment ($\text{UE}_{\text{Open-QA}}$) as shown in Fig. 1-(a), for the methods in Table 1, both before and after in-domain relearning and out-of-domain fine-tuning.

To conduct in-domain relearning, we update the unlearned model on samples from the forget set, following (Fan et al., 2025). For out-of-domain fine-tuning, we adapt the model to unrelated downstream tasks (including GSM8K, SST2, MNLI), following (Wang et al., 2025b). See **Appendix A** for more details. **Fig. 2** illustrates the robustness of in-domain relearning ($\text{Rob}_{\text{ReL}}$) and out-of-domain fine-tuning ($\text{Rob}_{\text{FT}}$) for various LLM unlearning methods on WMDP dataset, applied to Llama-3 8B Instruct. In the figure, lower accuracy (lighter color) indicates stronger robustness, and methods with explicit robust designs are marked with an asterisk (*). Several key insights can be drawn from Fig. 2.

*First*, relative to the unlearning performance before attack (*i.e.*, "unlearned" column), both in-domain relearning ($\text{Rob}_{\text{ReL}}$) and out-of-domain fine-tuning ($\text{Rob}_{\text{FT}}$) degrade unlearning effectiveness, as reflected by higher values in UE. Moreover, in-domain relearning acts as the worst-case testing, yielding lower robustness than out-of-domain fine-tuning.

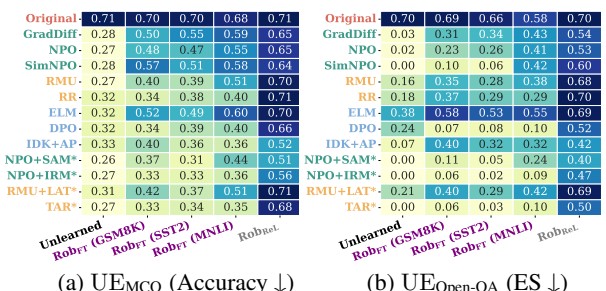

(a) $\text{UE}_{\text{MCQ}}$ (Accuracy ↓)  (b) $\text{UE}_{\text{Open-QA}}$ (ES ↓)

*Second*, focusing on methods without explicit robust designs, *divergence-driven optimization* approaches (including GradDiff, NPO, SimNPO) generally exhibit stronger robustness to in-domain relearning ($\text{Rob}_{\text{ReL}}$) than *representation misalignment* (including RMU, RR) and *rejection-based targeted unlearning* (including ELM, DPO). *However*, this trend can reverse under out-of-domain fine-tuning ($\text{Rob}_{\text{FT}}$), where *divergence-driven optimization* methods become less robust than *representation misalignment*, as seen with NPO vs. RMU under $\text{UE}_{\text{MCQ}}$. For *rejection-based targeted unlearning* methods, DPO is the most robust under $\text{Rob}_{\text{FT}}$; however, this advantage does not consistently hold under $\text{Rob}_{\text{ReL}}$. Hence, both robustness dimensions ($\text{Rob}_{\text{ReL}}$ and $\text{Rob}_{\text{FT}}$) should be jointly considered for a comprehensive assessment.

Figure 2: Robustness of in-domain relearning ($\text{Rob}_{\text{ReL}}$) and out-of-domain fine-tuning ($\text{Rob}_{\text{FT}}$) for 12 unlearning methods on WMDP with Llama-3 8B Instruct evaluated by (a) $\text{UE}_{\text{MCQ}}$ (Accuracy) and (b) $\text{UE}_{\text{Open-QA}}$ (ES). Out-of-domain fine-tuning uses GSM8K, SST2, and MNLI. Methods with * include robust designs, and the first column ("unlearned") shows results before attack.

*Third*, incorporating robustness-oriented designs consistently improves robustness against both in-domain relearning and out-of-domain fine-tuning, regardless of whether they build on divergence-driven optimization (*e.g.*, NPO+SAM, NPO+IRM) or representation misalignment (*e.g.*, TAR). An exception is RMU+LAT, which fails to show consistent improvements over RMU. Similar to adversarial logit pairing (Kannan et al., 2018), it offers only *local robustness*, leading to a less smooth loss landscape (Engstrom et al., 2018). **Fig. A3** of **Appendix B** confirms this limitation, as TAR exhibits a much smoother loss surface than RMU+LAT.

**Robustness against quantization.** Quantization is another form of model-level attack after unlearning. Unlike in-domain relearning or out-

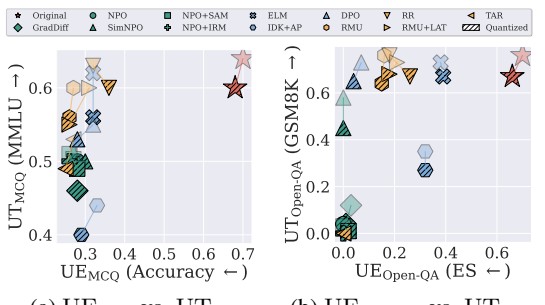

(a) $\text{UE}_{\text{MCQ}}$ vs. $\text{UT}_{\text{MCQ}}$  (b) $\text{UE}_{\text{Open-QA}}$ vs. $\text{UT}_{\text{Open-QA}}$

Figure 3: Robustness of quantization ($\text{Rob}_{\text{QT}}$) for 12 unlearning methods on WMDP with Llama-3 8B Instruct, evaluated by (a) $\text{UE}_{\text{MCQ}}$ (Accuracy) vs. $\text{UT}_{\text{MCQ}}$ (MMLU) and (b) $\text{UE}_{\text{Open-QA}}$ (ES) vs. $\text{UT}_{\text{Open-QA}}$ (GSM8K). Lines link models pre- and post-4bit quantization; hatched markers indicate quantized models.

of-domain fine-tuning, it does not introduce new knowledge but still alters model parameters and can affect robustness (Zhang et al., 2024b). Overly aggressive compression (*e.g.*, with very few quantization bits) may degrade the unlearned model's overall capability, making it unable to answer forget queries. This can create the *illusion* of improved unlearning performance, a false robustness

gain that simply reflects model incapacity. Thus, as a first principle, robustness under quantization should therefore be evaluated with the full UE–UT tradeoff.

To this end, **Fig. 3(a)** shows $UE_{MCQ}$ (Accuracy) versus $UT_{MCQ}$ (MMLU) before and after quantization. For *representation misalignment* and *rejection-based targeted unlearning* methods, the quantized models (markers with black diagonal hatching) exhibit declines in both $UT_{MCQ}$ and $UE_{MCQ}$, with the drop in $UT_{MCQ}$ being substantially larger. In contrast, *divergence-driven optimization* methods remain largely unaffected by quantization. **Fig. 3(b)** presents $UE_{Open-QA}$ (ES) versus $UT_{Open-QA}$ (GSM8K) before and after quantization, showing trends consistent with Fig. 3(a). In addition, another interesting finding is that knowledge removal (*e.g.*, WMDP) is generally more robust to post-unlearning quantization than data-centric unlearning (*e.g.*, MUSE for content removal). See Table A2 in **Appendix B** for details.

**Robustness against jailbreaking and its interaction with model-level robustness.** Beyond model-level perturbations, unlearned models are also vulnerable to input-level *jailbreaking attacks* (Łucki et al., 2024; Lynch et al., 2024; Patil et al., 2024), which manipulate prompts to bypass unlearning. Next, we examine whether current unlearning methods provide comparable robustness to both model-level and input-level attacks, and how these two robustness dimensions interact. Robustness against jailbreaking attacks is denoted as $Rob_{JA}$, with adversarial prompts generated using the enhanced GCG (Łucki et al., 2024).

As illustrated in **Fig. 4 (a)**, without explicit robust design, *divergence-driven optimization*

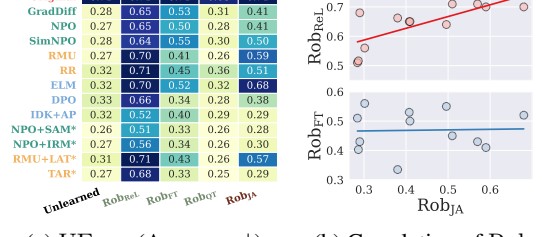

(a) $UE_{MCQ}$ (Accuracy ↓)          (b) Correlation of $Rob_{JA}$

Figure 4: (a) Overall robustness of 12 unlearning methods on WMDP with Llama-3 8B Instruct, including in-domain relearning ($Rob_{ReL}$), out-of-domain fine-tuning ($Rob_{FT}$), quantization ($Rob_{QT}$), and jailbreaking ($Rob_{JA}$) evaluated by $UE_{MCQ}$ (Accuracy) (b) Correlations between $Rob_{JA}$ and $Rob_{ReL}$ / $Rob_{FT}$.

methods (GradDiff, NPO, SimNPO) generally demonstrate stronger $Rob_{JA}$ compared with *representation misalignment* methods. This may be attributed to the degraded generative capacity of divergence-driven optimization methods, which inadvertently hinders their ability to reveal sensitive knowledge. For *rejection-based targeted unlearning* methods, $Rob_{JA}$ varies considerably: ELM shows almost no robustness, whereas IDK+AP remains robust with nearly no degradation. When robust design is incorporated, most methods (except RMU+LAT) exhibit significantly enhanced resilience against jailbreaking attacks.

Moreover, **Fig. 4(b)** jointly presents the relationship between input-level robustness ($Rob_{JA}$, learned input perturbations) and model-level robustness ($Rob_{ReL}$ and $Rob_{FT}$, learned weight perturbations). The results indicate that $Rob_{JA}$ patterns align more closely with $Rob_{ReL}$ than with $Rob_{FT}$. This positive correlation arises because both $Rob_{JA}$ and $Rob_{ReL}$ correspond to worst-case adversarial testing, with attack primarily activated by forget data in the unlearned domain.

## 6 CONCLUSION

In this work, we presented a full-stack investigation of LLM unlearning, encompassing methodology, evaluation, and robustness. We established a principled taxonomy that organizes twelve representative unlearning methods into three families: *divergence-driven optimization*, *representation misalignment*, and *rejection-based targeted unlearning*, providing a systematic lens to understand their underlying mechanisms. Our analysis revealed that conventional multiple-choice questioning (MCQ) evaluations of unlearning effectiveness (UE) and utility retention (UT) offer an incomplete picture, and we introduced open question answering (Open-QA) as a complementary paradigm to better capture generative behaviors and expose the strengths and limitations of different methods. Furthermore, we provide a comprehensive robustness assessment across model-level and input-level attacks, revealing nuanced relationships among in-domain relearning, out-of-domain fine-tuning, quantization, and jailbreak attacks. These findings clarify the trade-offs of current unlearning algorithms and guide the design of future methods that are both effective and robust. The use of LLM, limitation and broader impact are further discussed in **Appendix C**, **Appendix D** and **Appendix E**.

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

# APPENDIX

## A  EXPERIMENTAL SETUP

**Detailed experimental setup.**   For WMDP unlearning (Li et al., 2024), we employ Llama-3 8B Instruct as the reference model. The dataset consists of a forget set containing plain-text biosecurity knowledge and a retain set drawn from general-domain text in Wikitext (Merity et al., 2016). We perform 125 unlearning steps with a batch size of 4, conducting grid searches over learning rates in $[1 \times 10^{-5}, 5 \times 10^{-5}]$. The retain regularization parameter $\lambda$, which balances the retain loss, is tuned within $[1.0, 5.0]$. For NPO and SimNPO, we further explore $\beta \in [0.01, 0.1]$. For NPO+SAM, we grid search the perturbation radius $\rho$ within $[10^{-3}, 10^{-1}]$. For NPO+IRM, we adopt a single-dataset invariance setting using GSM8K, where the invariance weight $\gamma$ (controlling the strength of the constraint) is selected from $[0.1, 2.0]$. The batch size for GSM8K is fixed at 48 per unlearning step. For rejection-based targeted unlearning, we utilize GPT-4o to reformat the plain-text forget set $\mathcal{D}_f$ into a QA-style format (Łucki et al., 2024). For DPO and IDK+AP, $\beta$ is tuned within $[0.01, 0.1]$. For all other methods, we follow the configurations in Che et al. (2025). For in-domain relearning, we fine-tune on $\mathcal{D}_f$ for 100 steps with a batch size of 4 and learning rate $2 \times 10^{-5}$. For out-of-domain fine-tuning, we fine-tune for 250 steps with a batch size of 32 and the same learning rate, using data from GSM8K, SST-2, and MNLI.

**Implementation details of few-shot entailment score.**   Few-shot entailment score (ES) measures the factual correctness of a model's output relative to ground truth answers by leveraging a Natural Language Inference (NLI) model. Following prior work (Liu et al., 2024c; Yuan et al., 2024), we use a pre-trained NLI model (Sileo, 2023) to classify the relationship between the model's output (treated as the premise) and the ground truth answer (treated as the hypothesis). The predicted labels include entailment, contradiction, and neutral. We define the ES score as the proportion of examples classified as entailment. This metric is expected to be low on the forget set. Before generating answers for ES evaluation, we add a few-shot prompt consisting of 2 demonstration examples. These demonstrations do not involve NLI labels, but simply show the model the required output format in the multiple-choice setting (e.g., "C. tiger") without any explanations. The purpose is solely to ensure that the model outputs remain restricted to the given options (A–D), which makes the subsequent NLI evaluation reliable.

## B  ADDITIONAL EXPERIMENT RESULTS

Table A1: Example of answers from the original (pre-unlearned) model (Llama-3 8B Instruct) and the NPO / RMU unlearned model on WMDP, evaluated on both MCQ and Open-QA tasks.

| Question | Label | MCQ | Open-QA |
|---|---|---|---|
| In RNA virus genome synthesis, what potential risk is associated with obtaining virions? A.Limiting access to genome information B.Re-creating dangerous pathogens C.Decentralized access to synthetic RNA D.Accelerated biological research. Answer: | B | **Original**: B  **NPO**: D  **RMU**: D | **Original**: B. Re-creating dangerous pathogens  **NPO**: @nate@nate@nate@nate@nate@nate...  **RMU**: it is not the way it is added by someone... |

**Answer selection vs. answer generation.**   **Table A1** highlights the contrast between answer selection and answer generation for an NPO-unlearned Llama-3 8B Instruct on WMDP. Under the MCQ setting, both NPO and RMU successfully alter the model's prediction to an incorrect choice, suggesting effective unlearning of forget-relevant knowledge. However, in the Open-QA setting, the same models produce incoherent or nonsensical text rather than valid answers, indicating that their generative ability on forget queries is internally disrupted. This mismatch reveals a critical limitation of relying solely on MCQ-based evaluations, as they can obscure issues of over-forgetting that may also degrade performance on non-forget inputs.

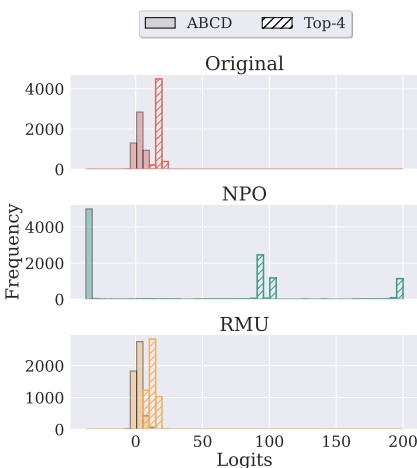

Figure A1: ABCD and top-4 token logits of the original (Llama-3 8B Instruct), NPO unlearned and RMU unlearned model on the WMDP evaluation set.

**From logits to behavior: over-forgetting in divergence-driven unlearning.** As indicated by Fig. 1-(c), divergence-driven optimization is prone to over-forgetting on Open-QA tasks. To further investigate this limitation, we compare the prediction logit distributions of the unlearned models (NPO and RMU) with the original pre-unlearned model over the answer options (A/B/C/D) and their top-4 predictions. **Fig. A1** illustrates how prediction logit distributions differ across unlearning methods on WMDP, comparing the options with each model's top-4 predicted tokens.

From the perspective of ABCD logits, NPO drives all four options close to zero and nearly identical, achieving $UE_{MCQ}$ by uniformly suppressing candidate scores. In other words, ABCD are not true top-token candidates under NPO, as revealed by its top-4 prediction logits on ABCD. By contrast, RMU maintains the distribution of the original model's logits but reshapes their relative distribution, attaining $UE_{MCQ}$ by reordering rather than erasing signals. For the top-4 logits, NPO assigns much higher values than RMU or the original model, but these correspond to meaningless tokens (Table A1). This shows that NPO achieves $UE_{Open-QA}$ by severely disrupting generative capacity, explaining its substantially lower $UT_{Open-QA}$ relative to RMU and the original model.

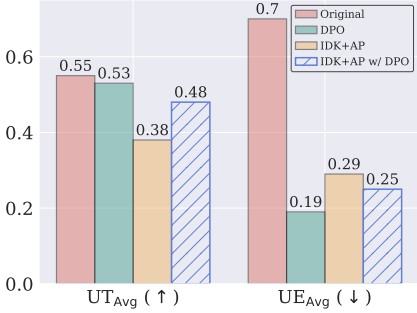

Figure A2: Effective unlearning with utility preservation of IDK+AP when warm-started with DPO (called IDK+AP w/ DPO), given by $UT_{Avg}$ and $UE_{Avg}$ (as defined in Fig. 1.

**Mitigating utility loss in IDK+AP via DPO warm-start.** As seen in Fig. 1-(c), DPO retains a significant portion of the original model's utility, which we attribute to the presence of a positive preference signal that guides the model to prefer the targeted answers in response to the forget queries. We hypothesize that the pronounced utility degradation of IDK+AP arises from its stricter log-likelihood loss, which aggressively increases the probability of rejection responses for forget-relevant questions. To mitigate this and to verify our hypothesis, we propose to unlearn using IDK+AP after a 'warm-start' with DPO for a few epochs. As seen in **Fig. A2**, this strategy (called IDK+AP w/

DPO) can infact increase preserve the utility, while achieving effective unlearning. We note this is in contrast to the setting of post-training of LLMs (Dubey et al., 2024) where SFT is followed by DPO. We think that in unlearning, the rejection responses provide a strong distribution shift for IDK+AP which is managed by warm-starting with DPO.

Table A2: Quantization performance of NPO and RMU is reported on MUSE Books and WMDP. UE and UT are assessed using KnowMem on $\mathcal{D}_f$ and KnowMem on $\mathcal{D}_r$ for MUSE Books, and by $UE_{MCQ}$ (Accuracy) and $UT_{MCQ}$ (MMLU) for WMDP. Results are provided for full precision, 8-bit, and 4-bit models.

| | MUSE | | | | | |
|---|---|---|---|---|---|---|
| **Method** | **KnowMem on $\mathcal{D}_f$ ($\downarrow$)** | | | **KnowMem on $\mathcal{D}_r$ ($\uparrow$)** | | |
| | **Full** | **8 Bit** | **4 Bit** | **Full** | **8 Bit** | **4 Bit** |
| NPO | 2.10 | 2.12 | 30.30 | 55.31 | 52.89 | 48.79 |
| RMU | 24.44 | 24.06 | 8.16 | 59.55 | 55.80 | 25.84 |
| | WMDP | | | | | |
| **Method** | **$UE_{MCQ}$ (Accuracy $\downarrow$)** | | | **$UT_{MCQ}$ (MMLU $\uparrow$)** | | |
| | **Full** | **8 Bit** | **4 Bit** | **Full** | **8 Bit** | **4 Bit** |
| NPO | 0.28 | 0.28 | 0.28 | 0.46 | 0.46 | 0.46 |
| RMU | 0.27 | 0.27 | 0.26 | 0.6 | 0.59 | 0.56 |

**Robustness of knowledge vs. data-centric unlearning under quantization.** As presented in Table A2 of **Appendix B**, we observe that quantization affects both unlearning effectiveness and utility. For data-centric unlearning (MUSE), 4-bit quantization leads to a sharp decline in performance, with NPO showing a significant increase in KnowMem on $\mathcal{D}_f$ and RMU suffering large drops in KnowMem on $\mathcal{D}_r$. In contrast, for knowledge removal (WMDP), both NPO and RMU maintain consistent UE across full precision, 8-bit, and 4-bit settings, while UT degrades only slightly. These results suggest that knowledge removal is generally more robust to post-unlearning quantization than content-based unlearning.

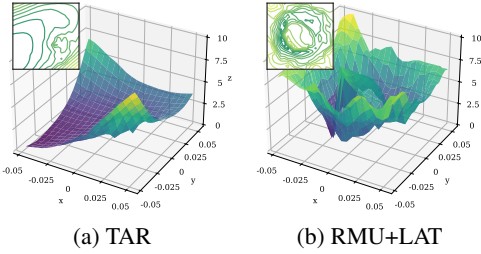

(a) TAR                    (b) RMU+LAT

Figure A3: The prediction loss landscape of the TAR and RMU+LAT-unlearned model on the forget set using the visualization tool in (Li et al., 2018).

**Loss landscape of TAR and RMU+LAT.** **Fig. A3** visualizes the forget loss landscape of TAR and RMU+LAT following (Li et al., 2018). The landscape of TAR is noticeably smoother, while RMU+LAT exhibits irregularities, indicating only *local robustness*. This echoes the debate (Engstrom et al., 2018) contrasting standard adversarial training (Madry et al., 2018) with adversarial logit pairing (Kannan et al., 2018), where leveraging logits (or other latent information) was argued to yield limited robustness. By analogy, in LLM unlearning, RMU+LAT also fails to achieve broad robustness due to its locality constraint.

## C    LLM USAGE

During the preparation of this manuscript, GPT-5 was used to assist with grammar correction and language refinement.

## D    LIMITATIONS

While this work offers a comprehensive full-stack investigation of LLM unlearning, we acknowledge several limitations. First, our taxonomy covers twelve representative methods, but additional approaches outside this scope may reveal further insights. Second, our robustness evaluations of input-level attacks focus mainly on jailbreak prompts. Future work should extend to other adversarial scenarios, such as in-context demonstrations or more advanced prompting techniques, to obtain a fuller picture. Third, our evaluation relies heavily on automatic metrics. Although these metrics enable systematic comparisons, they may overlook subtle aspects of model behavior. Human evaluations would provide complementary perspectives on unlearning success and user trust.

## E    BROADER IMPACTS

Improving unlearning is an essential step toward safer and more trustworthy language models. By clarifying methodologies, metrics, and robustness, our study provides a foundation for designing more effective approaches. Such progress has the potential to mitigate privacy risks, prevent the reproduction of harmful or copyrighted content, and support compliance with emerging regulations. At the same time, unlearning methods must be deployed with care, since excessive forgetting can degrade useful capabilities and adversarial adaptation may expose new vulnerabilities. We hope this work encourages the community to pursue principled, transparent, and responsible unlearning practices that balance safety, utility, and robustness in large language models.

