# OpenReview forum: "LLM Unlearning Under the Microscope: A Full-Stack View on Methods and Metrics"
_ICLR.cc/2026/Conference — Submitted to ICLR 2026_

### Official Review · Reviewer_VkyJ · 2025-10-27

**Soundness:** 2
**Presentation:** 2
**Contribution:** 2
**Rating:** 2
**Confidence:** 4

**Summary:**

This paper presents a comprehensive study of unlearning in large language models. The authors first propose a taxonomy that categorizes existing unlearning strategies into three main families: divergence-driven optimization, representation misalignment, and rejection-based targeted unlearning. They then show that common multiple-choice (MCQ) evaluations fail to reflect true model forgetting and utility retention, and introduce Open-QA–style metrics (such as entailment scoring) to capture generative behavior more faithfully. Finally, the paper conducts thorough robustness assessments under various post-unlearning attacks, including in-domain relearning, out-of-domain fine-tuning, quantization, and jailbreak prompts. The study concludes with insights on trade-offs between effectiveness, utility, and robustness, offering guidance for future unlearning designs.

**Strengths:**

**Strength 1**: The paper proposes a structured taxonomy of twelve recent LLM unlearning approaches, neatly grouping them into three methodological families. This taxonomy is well motivated and useful for organizing a fast-moving area with fragmented prior work.

**Strength 2**: The paper provides a structured and clear examination of robustness across multiple threat models, including model-level attacks (relearning, out-of-domain fine-tuning, quantization) and input-level jailbreak prompts. The visualizations in Figures 2–4 effectively highlight cross-method differences and make robustness trade-offs and vulnerabilities easy to interpret.

**Weaknesses:**

**Weakness 1**: The paper claims to provide a “full-stack” taxonomy and evaluation perspective on LLM unlearning. However, several directly relevant surveys and evaluation studies are missing in the related work and positioning, such as Wang et al.(2024) and Blanco-Justicia et al. (2025). Since the contribution of this work heavily relies on its claim to offer a uniquely systematic overview, the absence of these discussions weakens the novelty.

**Weakness 2**: Given the paper’s emphasis on distinguishing true unlearning from model degradation, the absence of standard behavioral diagnostics such as perplexity, KL divergence, or exact memorization weakens the evidence for its UE–UT trade-off claims. Incorporating at least one of these metrics would help confirm that observed performance changes reflect targeted forgetting rather than general model collapse.

**Weakness 3**: The main empirical claims are derived almost entirely from one benchmark (WMDP-Bio) and one model architecture (Llama-3-8B). Without broader validation across other standard unlearning settings (e.g., TOFU, RWKU) or larger-scale models, it remains unclear whether the reported method-family tradeoffs and robustness patterns generalize.

[1] Wang, Q., Han, B., Yang, P., Zhu, J., Liu, T., & Sugiyama, M. (2024). Towards effective evaluations and comparisons for llm unlearning methods. arXiv preprint arXiv:2406.09179.
[2] Blanco-Justicia, A., Jebreel, N., Manzanares-Salor, B., Sánchez, D., Domingo-Ferrer, J., Collell, G., & Eeik Tan, K. (2025). Digital forgetting in large language models: A survey of unlearning methods. Artificial Intelligence Review, 58(3), 90.

**Questions:**

Question 1: Are there plans or recommendations for extending the evaluation to other datasets, or to different LLM architectures? Any preliminary results would increase confidence in generality.

Question 2: Can the authors clarify the sampling strategy for selecting (x, y) pairs in NPO/SimNPO and RMU training? Specifically, are forget samples drawn uniformly or in a class-balanced manner, and are rare forget types specially treated? Clearer description would improve reproducibility and confidence in the fairness of comparisons.

Question 3: To better understand the reliability of the Open-QA evaluation, could the authors clarify how they verify the correctness of reference answers used in the entailment-score calculation, and whether this automatic metric shows reasonable agreement with human judgments, including any known cases where it may misjudge outputs?

---

> ### Author Response · Authors · 2025-11-22
> **Response to Reviewer VkyJ (Part 1)**
>
> **Q1.** Response on related work (Weakness 1)
>
> **A1.** Thank you for pointing out the two related works by Wang et al. [1] and Blanco-Justicia et al. [2]. We appreciate this and will explicitly include and discuss them in the revised version. Wang et al. propose an evaluation framework with a strong emphasis on metric design and calibration, while Blanco-Justicia et al. provide a survey of digital forgetting in LLMs.
>
> **However, our work differs from the prior work in our focus and contributions.**
>
> 1. Our goal is not only to evaluate but to provide a full-stack, methodology-guided analysis of LLM unlearning, from categorizing method families, to rethinking UE/UT evaluation, to analyzing robustness through the lens of these methodological principles and their relationships.
> 2. Novel contribution in a principled taxonomy of unlearning methods (Sec. 3). Neither Wang et al. nor Dorna et al. propose a principled categorization of stateful unlearning methods. Our Section 3 provides a taxonomy of twelve representative methods, grouped into: divergence-driven optimization, representation misalignment, and rejection-based targeted unlearning. This taxonomy enables and guides us to reason about why different methods behave differently under MCQ vs Open-QA, and why their robustness profiles diverge across model-level and input-level perturbations. For instance, rejection-based methods, which receive minimal attention in [1, 2], exhibit unusually strong Open-QA performance despite low MCQ performance (Lines 334-349).
> 3. Novel contribution on Open-QA as an important lens beyond MCQ (Sec. 4). While Wang et al. and Dorna et al. discuss evaluation faithfulness, neither performs a systematic MCQ vs Open-QA analysis grounded in method categories. Our work shows that: MCQ-based UE often overstates unlearning success, Open-QA reveals over-forgetting or utility loss not detectable by MCQ, and method families behave fundamentally differently under these two lenses.
> 4. Novel contribution on finer-level weight perturbations (in-domain vs. out-of-domain) and joint analysis of weight-level and input-level robustness (Sec. 5). Existing works evaluate robustness primarily as a metric property. Yet, we instead analyze method-level robustness by jointly examining: in-domain relearning, out-of-domain fine-tuning,  quantization, and input-level jailbreak attacks. And we connect these robustness dimensions back to the method families introduced in Sec. 3. This yields new insights, such as: divergence-driven optimization is typically more resilient to in-domain relearning; representation misalignment is more resistant to out-of-domain fine-tuning; jailbreak robustness aligns more closely with in-domain relearning than with out-of-domain fine-tuning. These cross-method, cross-metric insights are not explored in Wang et al. or Dorna et al.
> 5. Benchmarking choice on WMDP. Finally, we deliberately focus on WMDP and use it as our primary experimental benchmark because it does not require additional fine-tuning on the forget corpus prior to unlearning, thereby aligning more closely with real-world deployment where base models are used as-is. In contrast, benchmarks like TOFU and MUSE require fine-tuning on domain-specific corpora (e.g., synthetic authors or Harry Potter text) before unlearning, which can degrade general capabilities and introduce overfitting and domain shift, complicating fair unlearning assessment.
>
> As shown in **Table R1**, we report the utility of the original LLaMA-2-7B model and of models fine-tuned on TOFU and MUSE Books (which will be used for unlearning), following the settings of the original papers. We evaluate utility on GSM8K and IFEval, and observe that the TOFU- and MUSE-finetuned models perform substantially worse than the original LLaMA-2-7B. We were concerned that using such already-degraded models as the starting point for unlearning could create a false impression of “poor” utility after unlearning and an artificially inflated sense of “good” unlearning performance when presenting our insights.
>
> Table R1. GSM8K and IFEval of original and TOFU- and MUSE-finetuned LLaMA-2-7B
> | Method  | GSM8k | IFEval |
> |:-------:|:-----:|:------:|
> | Original| 0.23  |  0.44  |
> |  TOFU   | 0.10  |  0.13  |
> |  MUSE   | 0.08  |  0.24  |

---

> ### Author Response · Authors · 2025-11-22
> **Response to Reviewer VkyJ (Part 2)**
>
> **Q2.** Response on metrics choose (Weakness 2)
>
> **A2.** Thank you for this question. A large body of prior work (e.g., [1, 3–5]) has shown that these metrics are often **not reliable** indicators of successful unlearning:
>
> - **Perplexity (PPL):** While PPL is a standard metric for language modeling quality, in unlearning settings the *global* PPL of the model may change very little, even when the target knowledge has not been fully removed.
> - **KL divergence:** KL measures distributional differences, but in unlearning, the model may simply change its surface wording while still retaining the same underlying knowledge. In such cases, KL can look “improved” even though the sensitive or forgotten content is still implicitly present.
> - **Exact memorization:** This only checks whether the model can reproduce training data *verbatim*. Removing verbatim copies is not sufficient to guarantee forgetting: the model may still preserve the same knowledge via paraphrasing, partial summaries, or rephrased facts.
>
> For these reasons, we do not treat PPL, KL, or exact memorization as primary metrics in our study. In the original WMDP paper, UE and UT are evaluated only using MCQ-style metrics. However, as shown in Figure 1 of our paper, relying solely on MCQ metrics is insufficient to compare methods with fundamentally different behaviors. Therefore, we augment the evaluation with a suite of Open-QA metrics for UE (ES) and UT (like IFEval, GSM8K), which provide a more comprehensive and behaviorally grounded assessment.
>
>
> **Q3.** Response regarding additional models and datasets (Weakness 3 and Question 1)
>
> **A3.** Thank you for the valuable suggestion. In addition to our original experiments, we further evaluate on an alternative architecture and a larger-scale model, Qwen2.5-14B, and compare two representative families of methods: NPO, as a divergence-driven optimization method, and RMU, as a representation-misalignment method.
>
> For utility, we use MMLU and MathQA as $UT_{MCQ}$, and GSM8k and IFEval as $UT_{Open-QA}$. As shown in **Table R2**, even though NPO and RMU achieve almost similar $UT_{MCQ}$, RMU’s $UT_{Open-QA}$ is substantially higher than NPO’s. This aligns with our conclusion that “divergence-driven optimization methods often over-forget, breaking generation on forget queries, whereas representation-misalignment methods better preserve generation.”
>
> Table R2. Utility of NPO and RMU on unlearned Qwen2.5-14B.
>
> | Method | MMLU | MathQA | GSM8k | IFEval |
> |:------:|:----:|:------:|:-----:|:------:|
> |  NPO   | 0.72 |  0.51  | 0.00  |  0.14  |
> |  RMU   | 0.76 |  0.53  | 0.90  |  0.45  |
>
> For unlearning effectiveness, we use the accuracy on the WMDP evaluation set as $UE_{MCQ}$. We further evaluate robustness through (i) out-of-domain fine-tuning, (ii) in-domain relearning, and (iii) quantization, corresponding to the $Rob_{FT}$, $Rob_{Rel}$, and Quan columns, respectively. **Table R2** reports the results on Qwen2.5-14B and shows the same trends and insights as those analyzed in the main submission.
>
> From **Table R3**, we see that from the perspective of $UE_{MCQ}$, divergence-driven optimization (NPO) is typically more resilient to in-domain relearning (i.e., $Rob_{Rel}$), while representation-misalignment (RMU) better withstands out-of-domain fine-tuning (e.g., $Rob_{FT}(GSM8K)$).
>
> Table R3. $\text{UE}_\text{MCQ}$ and robustness of NPO and RMU on unlearned Qwen2.5-14B.
>
> | Method | Unlearned | $\text{Rob}_\text{FT}(\text{GSM8K})$ | $\text{Rob}_\text{FT}(\text{MNLI})$ | $\text{Rob}_\text{FT}(\text{SST2})$ | $\text{Rob}_\text{Rel}$ | Quan |
> |:------:|:---------:|:------------------------------------:|:-----------------------------------:|:-----------------------------------:|:-----------------------:|:----:|
> |  NPO   |   0.39    |                 0.60                 |                0.70                 |                0.53                 |          0.69           | 0.36 |
> |  RMU   |   0.30    |                 0.32                 |                0.35                 |                0.36                 |          0.79           | 0.29 |
>
> Last but not least, as we explained in our response to A1, our focus on the WMDP dataset is sufficient.

---

> ### Author Response · Authors · 2025-11-22
> **Response to Reviewer VkyJ (Part 3)**
>
> **Q4.** Response on sampling strategy in the training set (Question 2)
>
> **A4.** Thank you for the question. We clarify that in WMDP and in our unlearning setting, the model is not unlearning *classes* but unlearning *knowledge*. The forget set consists of hazardous biomedical facts (texts), not categorical labels, so the notion of “class-balanced” sampling does not apply. For NPO, SimNPO, and RMU, we randomly sample data from the WMDP dataset, as the existing benchmark did, and fix the random seed to ensure consistent sampling across runs.
>
> **Q5.** Response on entailment-score (Question 3)
>
> **A5.** Thank you very much for this careful reading and for raising this important point about the definition and robustness of the ES metric. **ES is always computed using the benchmark ground-truth annotation as the reference**, not the original model response.
>
> Concretely, for each Open-QA instance, we use an NLI model with
> - **premise** = model output (generation),
> - **hypothesis** = benchmark ground-truth answer (e.g., WMDP multiple-choice option),
>
> and define ES as the average binary entailment indicator over the evaluation set. The NLI backbone is the DeBERTa-v3 model from the original ES paper, instantiated as the HuggingFace checkpoint **`sileod/deberta-v3-base-tasksource-nli`**.
>
> To further demonstrate the validity of our Open-QA ES metric, we additionally conduct the following analyses.
>
> To further demonstrate the validity of our Open-QA ES metric, we additionally conduct the following analyses.
>
> 1. Agreement with human annotation on WMDP
>
>     We first compare Open-QA ES against human judgments on WMDP using Llama-3 8B Instruct. For each question, we obtain:
>     - a binary ES decision (“unlearned” vs. “not unlearned”)
>     - a binary human label.
>
>     This yields the following confusion matrix:
>     |                      | ES: Unlearn | ES: Not Unlearn |
>     |:--------------------:|:-----------:|:---------------:|
>     | **Human: Unlearn**   |    871      |       15        |
>     | **Human: Not Unlearn** |    20       |      367        |
>
>     The overall accuracy is $\text{Acc} = \frac{871 + 367}{871 + 15 + 20 + 367} = \frac{1238}{1273} \approx 0.9725,$ showing that ES agrees with human annotation **97.25%** of the time, i.e., ES is an extremely high-quality automatic judge. We further compute Cohen’s κ to account for agreement beyond chance. Cohen’s kappa is a statistic that quantifies how much two raters (or judges) agree when classifying items into categories, beyond what would be expected by chance [3].
>
>     Marginal counts:
>     - Human Unlearn = 886
>     - Human Not Unlearn = 387
>     - ES Unlearn = 891
>     - ES Not Unlearn = 382
>
>     The expected agreement under random matching is $p_e = \frac{886 \cdot 891 + 387 \cdot 382}{1273^2} \approx 0.578.$ The observed agreement is $p_o = 0.9725.$ Thus, $\kappa = \frac{p_o - p_e}{1 - p_e} = \frac{0.9725 - 0.578}{1 - 0.578} \approx 0.935.$ A κ of 0.935 falls into the **“almost perfect agreement”** regime in the Landis & Koch scale, indicating gold-standard level consistency between ES and human judgments.

---

> ### Author Response · Authors · 2025-11-22
> **Response to Reviewer VkyJ (Part 4)**
>
> 2. Multiple NLI backbones and confidence intervals
>
>     In our paper, ES is computed using the base NLI model `sileod/deberta-v3-base-tasksource-nli`. We further strengthen this analysis by:
>
>     1. adding a larger NLI model `sileod/deberta-v3-large-tasksource-nli`, and
>     2. introducing an LLM-as-judge variant(gpt-4o-mini) that directly decides entailment.
>
>     Because ES is the average of a binary entailment indicator over a finite evaluation set, its estimate is subject to sampling variability even when the generator and NLI backbone are deterministic. We therefore report **95% confidence intervals** using non-parametric bootstrap: for each model, we resample the $N$ evaluation examples with replacement $B = 1000$ replicates, recompute ES, and take the 2.5 and 97.5 percentile values. This captures uncertainty due to the finite benchmark size without assuming any parametric form.
>
>     The results are summarized in **Table R4**:
>
>     Table R4. ES of the original model and 12 unlearning methods on Llama-3 8B Instruct, evaluated using a base NLI model, a large NLI model, and an LLM-as-judge.
>
>     |   Method   | ES – base NLI |    95% CI     | ES – large NLI |    95% CI     | LLM as judge |
>     |:----------:|:-------------:|:-------------:|:--------------:|:-------------:|:-------------:|
>     |  Original  |     0.70      | [0.68, 0.73]  |      0.72      | [0.69, 0.74]  |     0.62      |
>     |  GradDiff  |     0.03      | [0.02, 0.04]  |      0.02      | [0.02, 0.03]  |     0.01      |
>     |    NPO     |     0.02      | [0.01, 0.03]  |      0.02      | [0.01, 0.03]  |     0.01      |
>     |   SimNPO   |     0.00      | [0.00, 0.00]  |      0.00      | [0.00, 0.00]  |     0.00      |
>     |  NPO+SAM   |     0.00      | [0.00, 0.00]  |      0.00      | [0.00, 0.00]  |     0.00      |
>     |  NPO+IRM   |     0.00      | [0.00, 0.00]  |      0.00      | [0.00, 0.00]  |     0.00      |
>     |    RMU     |     0.16      | [0.14, 0.18]  |      0.17      | [0.15, 0.19]  |     0.12      |
>     |     RR     |     0.18      | [0.16, 0.20]  |      0.20      | [0.18, 0.22]  |     0.15      |
>     |    ELM     |     0.38      | [0.35, 0.40]  |      0.40      | [0.37, 0.43]  |     0.31      |
>     |  RMU+LAT   |     0.21      | [0.19, 0.23]  |      0.21      | [0.19, 0.23]  |     0.17      |
>     |    TAR     |     0.00      | [0.00, 0.00]  |      0.00      | [0.00, 0.00]  |     0.00      |
>     |  IDK + AP  |     0.07      | [0.06, 0.09]  |      0.07      | [0.06, 0.09]  |     0.08      |
>     |    DPO     |     0.24      | [0.22, 0.27]  |      0.26      | [0.24, 0.29]  |     0.25      |
>
>     Across all three entailment backbones (base NLI, large NLI, and LLM-as-judge), the **relative conclusions are stable**. The confidence intervals are tight and do not alter the qualitative ranking of methods.
>
> > [1] Wang, Qizhou, et al. "Towards effective evaluations and comparisons for llm unlearning methods." arXiv preprint arXiv:2406.09179 (2024).
> >
> > [2] Blanco-Justicia, Alberto, et al. "Digital forgetting in large language models: A survey of unlearning methods." Artificial Intelligence Review 58.3 (2025): 90.
> >
> > [3] Feng, Zhili, et al. "Existing Large Language Model Unlearning Evaluations Are Inconclusive." arXiv preprint arXiv:2506.00688 (2025).
> >
> > [4] Ren, Jie, et al. "SoK: Machine Unlearning for Large Language Models." arXiv preprint arXiv:2506.09227 (2025).
> >
> > [5] Shi, Weijia, et al. "Muse: Machine unlearning six-way evaluation for language models." arXiv preprint arXiv:2407.06460 (2024).
> >
> > [6] Cohen, Jacob. "A coefficient of agreement for nominal scales." Educational and psychological measurement 20.1 (1960): 37-46.

---

> ### Comment · Reviewer_VkyJ · 2025-11-24
>
> I appreciate the authors for their thorough response, especially for the extra Qwen-14B experiments and human evaluation. These additions strengthen the paper, and I will raise my score.
>
> However, some of my main concerns still remain:
>
> 1. The decision not to evaluate on TOFU or other dataset prevents direct comparison with the most recent literature. The claim that base models are "degraded" during fine-tuning does not justify avoiding the field's most common benchmarks.
> 2. While the Open-QA insight is interesting, dropping standard metrics like PPL or KL for a custom protocol risks further dividing the evaluation landscape. To make a real impact, the proposed framework must connect with established metrics instead of completely replacing them.
>
> While the work has technical value, the absence of a standardized evaluation setup greatly limits its usefulness for the wider community. Also, the major weakness of the paper is the novelty. As in my original review and `Reviewer kNrs`, the taxonomy and the contribution in paper is not enough for the ICLR level. As a result, I will keep my review as reject

---

> > ### Author Response · Authors · 2025-11-25
> > **Additional response to Reviewer VkyJ.**
> >
> > Thank you again for your follow-up comments. We sincerely appreciate your reconsideration and the opportunity to further clarify our contributions. Below we address the remaining concerns.
> >
> > **1. On novelty and contribution**
> >
> > We respectfully disagree with the assessment that our work lacks sufficient novelty. This conclusion appears to overlook the core content of both our paper and our previous response. Our contributions are not limited to benchmarking; rather, we provide a methodology-driven analysis that introduces as detailed in **A1**. These listed contributions go substantively beyond prior work, and we respectfully ask the reviewer to assess their novelty and significance accordingly.
> >
> > **2. On the use of TOFU or other datasets**
> >
> > We also respectfully disagree with the claim that avoiding TOFU prevents meaningful comparison. As clarified earlier, TOFU-style settings require fine-tuning models on synthetic or domain-specific corpora, which substantially degrades the base model utility. As shown in **Table R1**, TOFU- and MUSE-finetuned LLaMA-2-7B models suffer severe degradation **even before** any unlearning is applied.
> > Using such already-degraded models as the starting point would create a false sense of unlearning success:
> >
> > > “We were concerned that using such already-degraded models as the starting point for unlearning could create a false impression of ‘poor’ utility after unlearning and an artificially inflated sense of ‘good’ unlearning performance.”
> >
> > Our choice of WMDP ensures that evaluation reflects genuine, behaviorally meaningful unlearning rather than artifacts of prior degradation.
> >
> > **3. On dropping PPL/KL and the need for established metrics**
> >
> > Regarding the concern that moving beyond PPL or KL “divides” the evaluation landscape, we respectfully disagree. As answered in **A2**, both metrics have been repeatedly shown to provide **only indirect evidence**:
> >
> > - **Perplexity (PPL):** global PPL can remain nearly unchanged even when the target knowledge is fully retained.
> > - **KL divergence:** KL may “improve” simply because the model rephrases the forgotten knowledge, despite retaining the underlying fact.
> >
> > For unlearning to have real impact and scientific clarity, the field needs **behavioral metrics** that directly assess whether the knowledge has been removed, not proxies that measure surface-level linguistic similarity. Our Open-QA framework provides such direct evidence,  as shown in **A5**.

---

> > > ### Comment · Reviewer_VkyJ · 2025-11-25
> > >
> > > I respectfully disagree with some author response and I will see the other review rating and decide to whether further need change my rating. As a result, I will keep my score as weak reject.

---

> > > > ### Author Response · Authors · 2025-11-25
> > > >
> > > > Dear Reviewer VkyJ,
> > > >
> > > > Thank you very much for your follow-up comment and for sharing your perspective. **We fully respect your views**, even if they differ from ours. We sincerely hope that, upon considering the full set of reviews and our detailed clarifications, you may find that our work introduces meaningful novelty and offers insights that extend beyond the existing literature. Regardless of the final rating, we genuinely appreciate your time, thoughtful engagement, and constructive feedback throughout the review process.
> > > >
> > > > Authors,

---

> > > > > ### Comment · Reviewer_VkyJ · 2025-11-25
> > > > >
> > > > > Thanks author response. I am sorry even I cost several hours reading the your paper and rebuttal again and again, I still think the novelty is limited. I think it more looks like a benchmark paper and I am fine with this track, even I suggest you transfer it to benchmark track. Personally, I also do the benchmark and I believe the original draft's experiment is so limited. Even after your rebuttal, the experiment set is suitable for an benchmark rate. I think the current score is my final rating if no accidence and I am willing reduce my confidence score to 3.

---

> > > > > > ### Author Response · Authors · 2025-11-26
> > > > > >
> > > > > > Thank you very much for taking the time to read our paper and rebuttal carefully. We respect your assessment and your view that the contribution appears closer to a benchmark-style paper. Yet, we still want to highlight that our intention goes beyond providing a benchmark. While we do evaluate multiple unlearning methods, this is done to support the broader methodological principles we summarize. The key insights presented in Sections 4 and 5 (see their summary boxes) are intended to uncover conceptual gaps in current LLM unlearning, rather than simply report performance across methods.
> > > > > >
> > > > > > We appreciate your honest feedback and understand your perspective. We value your suggestion regarding the benchmark track and your willingness to adjust the confidence score.
> > > > > >
> > > > > > Thank you again for your thoughtful engagement throughout the review process.

---

### Official Review · Reviewer_dRnP · 2025-10-27

**Soundness:** 4
**Presentation:** 4
**Contribution:** 2
**Rating:** 4
**Confidence:** 4

**Summary:**

This paper presents a comprehensive analysis of llm unlearning, offering a full-stack perspective that integrates methodological categorization, evaluation, and robustness. The authors identify the fragmentation and inconsistency in prior unlearning research and propose a taxonomy of twelve stateful unlearning methods.

The paper further reevaluates unlearning effectiveness (UE) and utility retention (UT), highlighting that popular benchmarks, such as WMDP, rely heavily on multiple-choice question (MCQ) format—an approach that overlooks the generative capabilities of LLMs. To address this, the authors advocate introducing open question answering (Open-QA) as a complementary evaluation lens. Open-QA captures the model’s actual generation ability and exposes hidden trade-offs, such as over-forgetting in divergence-driven methods that degrade utility beyond MCQ assessments.

The work also examines robustness against both model-level attacks (e.g., in-domain relearning, out-of-domain fine-tuning, quantization) and input-level attacks (e.g., prompt jailbreaking). Results reveal nuanced vulnerabilities: divergence-driven optimization methods resist relearning attacks better, whereas representation misalignment methods handle out-of-domain fine-tuning more robustly. The paper further shows that combining unlearning with robust optimization frameworks (e.g., SAM, IRM) improves resilience across threat types.

**Strengths:**

The authors are obviously the experts in the field. The draft is well structured, insightful, and easy to read. The paper is a concrete and precise summary of the current progress.

Some statements are interesting, such as divergence-driven optimization methods are typically more resilient to in-domain relearning, while representation misalignment methods better withstand out-of-domain fine-tuning.

**Weaknesses:**

As the first method to address unlearning and retention trade-off, I think GRU [1] need to be mentioned in Sec 3.

[1] GRU: Mitigating the Trade-off between Unlearning and Retention for Large Language Models

It somehow likes a surveying paper, maybe it is useful to discuss its difference and uniqueness with previous works, like [2].

[2] Rethinking Machine Unlearning for Large Language Models

Metrics are proposed beyond MCQ and Open-QA, such as those test the likelihood in generating the original responses. Forgive me if I made some mistakes, [3] suggests many metrics of this kind and states ES (same name as in this paper but actually different formulation) as a robust and reliable metric. I do not know if it should be mentioned for the overall concreteness. Also, Open-QA seems not new, but it seems that the authors  do not further sufficient references.


[3] OpenUnlearning: Accelerating LLM Unlearning via Unified Benchmarking of Methods and Metrics

A general problem I want to discuss with the authors is that, under the current common unlearning setup, do you think WMDP is strictly reliable? It seems that we will typically implicitly assume that the knowledge has been parameterized into the model, so, can we ensure that the knowledge in WMDP will exist in the considered models?

I cannot guarantee it is correct, but some researchers in this field told me RMU only works well for WMDP while being hard to tuning in achieving proper performance for other benchmarks. Therefore, the claim “representation misalignment generally outperforms rejection-based targeted unlearning“ is too strong and the conclusion is only constrained in the considered experimental setups.

Some paper discussed the drawbacks of divergence-based optimization methods in wrong reweighting [4], maybe it can be mentioned as a simple explanation about why over-forgetting. [4] also mentioned the drawback of rejection based method, stating something like mapping to new targets does not mean old knowledge is overwrite. Not sure if they should be mentioned for concreteness.

[4] Rethinking LLM Unlearning Objectives: A Gradient Perspective and Go Beyond

Personally, I think cross-language attack is also interesting, as it can reflect if the underlying knowledge has been removed or we just make the model to refuse outputting particular form of responses. Do the authors think it should be mentioned?


Maybe some attentions are required to explain why this paper is useful in more directly advancing the field. As a very bias and personal understanding about the tastes of the current community, open challenges as well as some potential solutions (and preliminary verifications) are required for acceptance. It is just a minor suggestion, I do not expect that the authors will come up with some new methodologies during rebuttal, but at least a summary of the current open question is required. Overall, this is a good paper that summarizes the current progress, I will definitely change my scores when the authors address my concerns.

**Questions:**

Kindly please see the drawbacks above.

---

> ### Author Response · Authors · 2025-11-22
> **Response to Reviewer dRnP (Part 1)**
>
> **Q1.** As the first method to address unlearning and retention trade-off, I think GRU [1] need to be mentioned in Sec 3.
> > [1] GRU: Mitigating the Trade-off between Unlearning and Retention for Large Language Models
>
> It somehow likes a surveying paper, maybe it is useful to discuss its difference and uniqueness with previous works, like [2].
> > [2] Rethinking Machine Unlearning for Large Language Models
>
> Metrics are proposed beyond MCQ and Open-QA, such as those test the likelihood in generating the original responses. Forgive me if I made some mistakes, [3] suggests many metrics of this kind and states ES (same name as in this paper but actually different formulation) as a robust and reliable metric.
> > [3] OpenUnlearning: Accelerating LLM Unlearning via Unified Benchmarking of Methods and Metrics
>
> Some paper discussed the drawbacks of divergence-based optimization methods in wrong reweighting [4], maybe it can be mentioned as a simple explanation about why over-forgetting. [4] also mentioned the drawback of rejection based method, stating something like mapping to new targets does not mean old knowledge is overwrite. Not sure if they should be mentioned for concreteness.
> > [4] Rethinking LLM Unlearning Objectives: A Gradient Perspective and Go Beyond
>
>
> **A1.** Thank you very much for pointing us to references [1–4]. We will revise the related work to more clearly position our contribution relative to these papers.
>
> Briefly, GRU [1] proposes a new unlearning objective that explicitly mitigates the trade-off between unlearning and retention and demonstrates its effectiveness across several benchmarks, so it is primarily an algorithmic contribution. Rethinking Machine Unlearning for LLMs [2] provides a broad conceptual and methodological overview of LLM unlearning, including scope, data–model interactions and high-level assessment principles, and is closer in spirit to a survey. OpenUnlearning [3] provides a standardized platform that integrates multiple unlearning methods and evaluation metrics, and further assesses metric faithfulness and robustness through stress tests, with a primary focus on the TOFU benchmark. Rethinking LLM Unlearning Objectives [4] develops the G-effect toolkit to analyze existing unlearning objectives from a gradient perspective and to diagnose their failure modes, including issues with divergence-based and rejection-based objectives.
>
> **However, our work differs from the prior work in our focus and contributions.**
>
> 1. Our goal is not only to evaluate but to provide a full-stack, methodology-guided analysis of LLM unlearning, from categorizing method families, to rethinking UE/UT evaluation, to analyzing robustness through the lens of these methodological principles and their relationships.
> 2. Novel contribution in a principled taxonomy of unlearning methods (Sec. 3). Neither Wang et al. nor Dorna et al. propose a principled categorization of stateful unlearning methods. Our Section 3 provides a taxonomy of twelve representative methods, grouped into: divergence-driven optimization, representation misalignment, and rejection-based targeted unlearning. This taxonomy enables and guides us to reason about why different methods behave differently under MCQ vs Open-QA, and why their robustness profiles diverge across model-level and input-level perturbations. For instance, rejection-based methods, which receive minimal attention in [1, 2], exhibit unusually strong Open-QA performance despite low MCQ performance (Lines 334-349).
> 3. Novel contribution on Open-QA as an important lens beyond MCQ (Sec. 4). While Wang et al. and Dorna et al. discuss evaluation faithfulness, neither performs a systematic MCQ vs Open-QA analysis grounded in method categories. Our work shows that: MCQ-based UE often overstates unlearning success, Open-QA reveals over-forgetting or utility loss not detectable by MCQ, and method families behave fundamentally differently under these two lenses.
> 4. Novel contribution on finer-level weight perturbations (in-domain vs. out-of-domain) and joint analysis of weight-level and input-level robustness (Sec. 5). Existing works evaluate robustness primarily as a metric property. Yet, we instead analyze method-level robustness by jointly examining: in-domain relearning, out-of-domain fine-tuning,  quantization, and input-level jailbreak attacks. And we connect these robustness dimensions back to the method families introduced in Sec. 3. This yields new insights, such as: divergence-driven optimization is typically more resilient to in-domain relearning; representation misalignment is more resistant to out-of-domain fine-tuning; jailbreak robustness aligns more closely with in-domain relearning than with out-of-domain fine-tuning. These cross-method, cross-metric insights are not explored in Wang et al. or Dorna et al.

---

> ### Author Response · Authors · 2025-11-22
> **Response to Reviewer dRnP (Part 2)**
>
> **Q2.** A general problem I want to discuss with the authors is that, under the current common unlearning setup, do you think WMDP is strictly reliable? It seems that we will typically implicitly assume that the knowledge has been parameterized into the model, so, can we ensure that the knowledge in WMDP will exist in the considered models? I cannot guarantee it is correct, but some researchers in this field told me RMU only works well for WMDP while being hard to tuning in achieving proper performance for other benchmarks. Therefore, the claim “representation misalignment generally outperforms rejection-based targeted unlearning“ is too strong and the conclusion is only constrained in the considered experimental setups.
>
> **A2.** We deliberately focus on WMDP and use it as our primary experimental benchmark because it does not require additional fine-tuning on the forget corpus prior to unlearning, thereby aligning more closely with real-world deployment where base models are used as-is. In contrast, benchmarks like TOFU and MUSE require fine-tuning on domain-specific corpora (e.g., synthetic authors or Harry Potter text) before unlearning, which can degrade general capabilities and introduce overfitting and domain shift, complicating fair unlearning assessment.
>
> As shown in **Table R1**, we report the utility of the original LLaMA-2-7B model and of models fine-tuned on TOFU and MUSE Books (which will be used for unlearning), following the settings of the original papers. We evaluate utility on GSM8K and IFEval, and observe that the TOFU- and MUSE-finetuned models perform substantially worse than the original LLaMA-2-7B. We were concerned that using such already-degraded models as the starting point for unlearning could create a false impression of “poor” utility after unlearning and an artificially inflated sense of “good” unlearning performance when presenting our insights.
>
> Table R1. GSM8K and IFEval of original and TOFU- and MUSE-finetuned LLaMA-2-7B
>
> | Method  | GSM8k | IFEval |
> |:-------:|:-----:|:------:|
> | Original| 0.23  |  0.44  |
> |  TOFU   | 0.10  |  0.13  |
> |  MUSE   | 0.08  |  0.24  |
>
> By contrast, the knowledge in WMDP has been present in the original model. For instance, the original Llama-3-8B-Instruct model achieves 72 percent accuracy on WMDP, which is well above the 25 percent random-guess baseline. This suggests that the benchmarked knowledge is meaningfully represented in the model parameters.
>
> In addition, RMU also demonstrates strong performance on additional benchmarks. **Table 3** in [3] and **Table D1** in [5] both indicate that RMU performs competitively on TOFU and MUSE, and in these settings, it consistently outperforms NPO.
>
> Furthermore,  prior work has predominantly relied on the utility metric UT_MCQ. As summarized in Table 1 of our paper, this can sometimes create the impression that various unlearning methods achieve similar levels of utility. When utility is evaluated through UT_Open-QA instead, representation-misalignment methods (like RMU) exhibit substantially better utility retention. We will make this distinction clearer in the revised version.

---

> ### Author Response · Authors · 2025-11-22
> **Response to Reviewer dRnP (Part 3)**
>
> **Q3.** Personally, I think cross-language attack is also interesting, as it can reflect if the underlying knowledge has been removed or we just make the model to refuse outputting particular form of responses. Do the authors think it should be mentioned?
>
> **A3.** We agree that cross-language attacks are a promising direction for probing whether the underlying knowledge has truly been removed, as opposed to merely suppressing specific surface forms in one language. However, in this paper we focus on **LLaMA-based checkpoints**, whose official documentation states that Llama 3 is primarily intended for **English** usage and that use in other languages is out of scope of the license and safety guarantees. Under this constraint, a rigorous cross-language evaluation would require more capable  models beyond the intended use of the released checkpoints.
>
> We will explicitly mention this limitation and briefly discuss cross-language attacks as a promising future direction in the revised version.
>
> > [1] Wang, Yue, et al. "GRU: Mitigating the Trade-off between Unlearning and Retention for LLMs." arXiv preprint arXiv:2503.09117 (2025).
> >
> > [2] Liu, Sijia, et al. "Rethinking machine unlearning for large language models." Nature Machine Intelligence (2025): 1-14.
> >
> > [3] Dorna, Vineeth, et al. "OpenUnlearning: Accelerating LLM Unlearning via Unified Benchmarking of Methods and Metrics." arXiv preprint arXiv:2506.12618 (2025).
> >
> > [4] Wang, Qizhou, et al. "Rethinking llm unlearning objectives: A gradient perspective and go beyond." arXiv preprint arXiv:2502.19301 (2025).
> >
> > [5] Pal, Soumyadeep, et al. "Llm unlearning reveals a stronger-than-expected coreset effect in current benchmarks." arXiv preprint arXiv:2504.10185 (2025).

---

> > ### Comment · Reviewer_dRnP · 2025-11-24
> >
> > The authors' feedback is decent and clear, which addresses much of my concerns. Accordingly, I increase my score to 8. Thanks and good luck!

---

> ### Author Response · Authors · 2025-11-24
> **Thank you very much for raising the score to 8**
>
> Dear Reviewer dRnP,
>
> Thank you again for your thoughtful comments and for taking the time to revisit our response. We are grateful that our clarifications were helpful, and we sincerely appreciate your revised score and kind encouragement.
>
> Thank you once more for your support and engagement.
>
> Warm regards,
>
> The Authors

---

### Official Review · Reviewer_8HQQ · 2025-11-03

**Soundness:** 3
**Presentation:** 3
**Contribution:** 2
**Rating:** 4
**Confidence:** 4

**Summary:**

This paper presents a “full-stack” re-examination of LLM unlearning. It organizes twelve representative methods into three families—(i) divergence-driven optimization (e.g., NPO, SimNPO, robustified with SAM/IRM), (ii) representation misalignment (e.g., RMU, RR, TAR/LAT), and (iii) rejection-based targeted unlearning (e.g., ELM, DPO, IDK+AP)—and proposes a unified evaluation protocol spanning multiple-choice (MCQ) and open-ended generation (OpenQA). Unlearning effectiveness $\mathrm{UE}$ and utility retention $\mathrm{UT}$ are defined for both regimes; for OpenQA, outputs are judged via an NLI-based entailment score (ES) against a reference answer. Aggregate views $\mathrm{UE}\_{\mathrm{Avg}}$ and $\mathrm{UT}\_{\mathrm{Avg}}$ summarize overall behavior.

On WMDP (notably WMDP-Bio) with Llama-3-8B-Instruct as the base model, the study highlights a systematic gap: divergence-driven methods can match representation-misalignment methods on $\mathrm{UE}\_{\mathrm{MCQ}}$ yet suffer marked drops in $\mathrm{UT}\_{\mathrm{OpenQA}}$, revealing over-forgetting that MCQ alone obscures. Robustness is examined under in-domain relearning, out-of-domain fine-tuning, post-unlearning quantization and jailbreaks; a practical recipe shows a DPO warm-start mitigating the utility collapse of IDK+AP.

**Strengths:**

The work tightly couples method taxonomy, metric design, and robustness analysis in one coherent framework. Pairing MCQ and OpenQA via $\mathrm{UE}$ and $\mathrm{UT}$ surfaces failure modes that MCQ alone cannot detect; the NLI-based ES moves the evaluation closer to realistic generative behavior.


The WMDP findings are crisply interpreted: divergence-driven methods may excel on $\mathrm{UE}\_{\mathrm{MCQ}}$ yet degrade $\mathrm{UT}\_{\mathrm{OpenQA}}$, and the logits inspection provides a concrete mechanism rather than post-hoc speculation. Robustness is thoughtfully scoped—separating in-domain relearning from out-of-domain fine-tuning, diagnosing quantization-induced capacity loss, and linking jailbreak susceptibility to training choices. The DPO warm-start for IDK+AP is practical and immediately adoptable without architectural changes.

I believe the most significant advantage of this paper lies in integrating the robustness of the model layer (intra-domain relearning, out-domain fine-tuning, quantization) with the input layer (jailbreaking) into a unified evaluation framework.

**Weaknesses:**

Headline trends are shown on a single base model (Llama-3-8B-Instruct). Whether the family-level narrative (e.g., representation-misalignment methods being more resilient to out-of-domain fine-tuning; divergence-driven methods being more vulnerable in OpenQA) holds across architectures (Gemma/Qwen) and scales (2B/14B) remains uncertain. Some studies have pointed out that different model sizes have certain influences and patterns on unlearning.

The OpenQA ES hinges on two choices under-specified in the main text: the reference answer policy (dataset ground truth vs. the original pre-unlearned model) and the NLI judge. Either can shift $\mathrm{UE}\_{\mathrm{OpenQA}}$ and $\mathrm{UT}\_{\mathrm{OpenQA}}$ and potentially flip method rankings; scorer variance/calibration and human agreement are not reported.

Robustness coverage, though broad, uses a narrow jailbreak family and a limited set of quantization bit-widths/schemes. I think that having a unified "budget" that coordinates computational cost, number of steps, and hyperparameter search range across different methods would help improve fairness and reproducibility across algorithm families.

**Questions:**

1) Can the core trends—MCQ/OpenQA divergence in the $\mathrm{UE}$–$\mathrm{UT}$ trade-off, family-specific robustness profiles, and the NPO logits-flattening effect—be reproduced on at least two **additional architectures** (e.g., Gemma/Qwen) and **different scale** (e.g., 2B and 13B)? A compact architecture × size matrix would establish external validity.


2) What puzzles me is that in Appendix A (Experimental Setup), the authors explicitly state that ES uses NLI to determine the implication relationship between the model output and the ground truth (benchmark annotation) (premise = model output, hypothesis = benchmark answer) and scores accordingly. However, in Section 4 of the main text (Open-QA as an important perspective for UE/UT evaluation), it states that ES measures the consistency between the model output and the original (unforgotten) model response (i.e., the "correct answer"). This is because if the original model output is considered the "correct answer," then when the original model itself is flawed, ES will incorrectly classify the output that "corrects towards the truth" as wrong; using the benchmark annotation as a reference will not. Since one of the most important arguments in this paper is that "Divergence-driven approaches are more prone to over-forgetting in Open-QA and representation misalignment is more stable," and **these conclusions rely on the ES metric of Open-QA**, if switching the reference source leads to a reversal in UE/UT rankings, the robustness of the main conclusions will be compromised. How should this conflict be handled? Could you report the consistency of conclusions reached under the two references? If possible, use multiple NLI judges (≥2) and a human-labeled subset to estimate ES variance/calibration and provide CIs for $\mathrm{UE}\_{\mathrm{OpenQA}}$ and $\mathrm{UT}\_{\mathrm{OpenQA}}$.


3) Extend bit-widths and include multiple schemes (GPTQ/AWQ/RTN). Plot full UE–UT curves to disentangle genuine robustness from capacity collapse, and define an explicit “unusable” region (e.g., $\mathrm{UT}\_{\mathrm{OpenQA}}$ below a percentile threshold) to prevent misleading gains in $\mathrm{UE}$ from capacity loss.

If you can clarify the above issues and supplement the experimental evidence to demonstrate the generalizability of the conclusions, I would be happy to raise score.

---

> ### Author Response · Authors · 2025-11-22
> **Response to Reviewer 8HQQ (Part 1)**
>
> **Q1.** Can the core trends—MCQ/OpenQA divergence in the UE-UT trade-off, family-specific robustness profiles, and the NPO logits-flattening effect—be reproduced on at least two additional architectures (e.g., Gemma/Qwen) and different scale (e.g., 2B and 13B)?
>
> **A1.** Thank you for the valuable suggestion. In addition to our original experiments, we further evaluate on an alternative architecture and a larger-scale model, Qwen2.5-14B, and compare two representative families of methods: NPO, as a divergence-driven optimization method, and RMU, as a representation-misalignment method.
>
> For utility, we use MMLU and MathQA as $UT_{MCQ}$, and GSM8k and IFEval as $UT_{Open-QA}$. As shown in **Table R1**, even though NPO and RMU achieve almost similar $UT_{MCQ}$, RMU’s $UT_{Open-QA}$ is substantially higher than NPO’s. This aligns with our conclusion that “divergence-driven optimization methods often over-forget, breaking generation on forget queries, whereas representation-misalignment methods better preserve generation.”
>
> Table R1. Utility of NPO and RMU on unlearned Qwen2.5-14B
>
> | Method | MMLU | MathQA | GSM8k | IFEval |
> |:------:|:----:|:------:|:-----:|:------:|
> |  NPO   | 0.72 |  0.51  | 0.00  |  0.14  |
> |  RMU   | 0.76 |  0.53  | 0.90  |  0.45  |
>
> For unlearning effectiveness, we use the accuracy on the WMDP evaluation set as $UE_{MCQ}$. We further evaluate robustness through (i) out-of-domain fine-tuning, (ii) in-domain relearning, and (iii) quantization, corresponding to the $Rob_{FT}$, $Rob_{Rel}$, and Quan columns, respectively. **Table R2** reports the results on Qwen2.5-14B and shows the same trends and insights as those analyzed in the main submission.
>
> From **Table R2**, we see that from the perspective of $UE_{MCQ}$, divergence-driven optimization (NPO) is typically more resilient to in-domain relearning (i.e., $Rob_{Rel}$), while representation-misalignment (RMU) better withstands out-of-domain fine-tuning (e.g., $Rob_{FT}(GSM8K)$).
>
> Table R2. $\text{UE}_\text{MCQ}$ and robustness of NPO and RMU on unlearned Qwen2.5-14B
>
> | Method | Unlearned | $\text{Rob}_\text{FT}(\text{GSM8K})$ | $\text{Rob}_\text{FT}(\text{MNLI})$ | $\text{Rob}_\text{FT}(\text{SST2})$ | $\text{Rob}_\text{Rel}$ | Quan |
> |:------:|:---------:|:------------------------------------:|:-----------------------------------:|:-----------------------------------:|:-----------------------:|:----:|
> |  NPO   |   0.39    |                 0.60                 |                0.70                 |                0.53                 |          0.69           | 0.36 |
> |  RMU   |   0.30    |                 0.32                 |                0.35                 |                0.36                 |          0.79           | 0.29 |

---

> ### Author Response · Authors · 2025-11-22
> **Response to Reviewer 8HQQ (Part 2)**
>
> **Q2**. Since one of the most important arguments in this paper is that "Divergence-driven approaches are more prone to over-forgetting in Open-QA and representation misalignment is more stable," and these conclusions rely on the ES metric of Open-QA, if switching the reference source leads to a reversal in UE/UT rankings, the robustness of the main conclusions will be compromised. How should this conflict be handled? Could you report the consistency of conclusions reached under the two references? If possible, use multiple NLI judges (≥2) and a human-labeled subset to estimate ES variance/calibration and provide CIs.
>
> **A2.** Thank you very much for this careful reading and for raising this important point about the definition and robustness of the ES metric. We apologize for the confusing wording between Appendix A and Section 4. In our implementation, **ES is always computed using the benchmark ground-truth annotation as the reference**, not the original model response.
>
> Concretely, for each Open-QA instance we use an NLI model with
> - **premise** = model output (generation),
> - **hypothesis** = benchmark ground-truth answer (e.g., WMDP multiple-choice option),
>
> and define ES as the average binary entailment indicator over the evaluation set. The NLI backbone is the DeBERTa-v3 model from the original ES paper, instantiated as the HuggingFace checkpoint **`sileod/deberta-v3-base-tasksource-nli`**.
>
> In the revised version, we will (i) explicitly state in Section 4 that ES is defined with respect to benchmark ground truth, and (ii) add a short remark explaining why we intentionally avoid using the original model response as the reference for Open-QA.
>
> To further demonstrate the validity of our Open-QA ES metric, we additionally conduct the following analyses.
>
> 1. Agreement with human annotation on WMDP
>
>     We first compare Open-QA ES against human judgments on WMDP using Llama-3 8B Instruct. For each question, we obtain:
>     - a binary ES decision (“unlearned” vs. “not unlearned”)
>     - a binary human label.
>
>     This yields the following confusion matrix:
>     |                      | ES: Unlearn | ES: Not Unlearn |
>     |:--------------------:|:-----------:|:---------------:|
>     | **Human: Unlearn**   |    871      |       15        |
>     | **Human: Not Unlearn** |    20       |      367        |
>
>     The overall accuracy is $\text{Acc} = \frac{871 + 367}{871 + 15 + 20 + 367} = \frac{1238}{1273} \approx 0.9725,$ showing that ES agrees with human annotation **97.25%** of the time, i.e., ES is an extremely high-quality automatic judge. We further compute Cohen’s κ to account for agreement beyond chance. Cohen’s kappa is a statistic that quantifies how much two raters (or judges) agree when classifying items into categories, beyond what would be expected by chance [3].
>
>     Marginal counts:
>     - Human Unlearn = 886
>     - Human Not Unlearn = 387
>     - ES Unlearn = 891
>     - ES Not Unlearn = 382
>
>     The expected agreement under random matching is $p_e = \frac{886 \cdot 891 + 387 \cdot 382}{1273^2} \approx 0.578.$ The observed agreement is $p_o = 0.9725.$ Thus, $\kappa = \frac{p_o - p_e}{1 - p_e} = \frac{0.9725 - 0.578}{1 - 0.578} \approx 0.935.$ A κ of 0.935 falls into the **“almost perfect agreement”** regime in the Landis & Koch scale, indicating gold-standard level consistency between ES and human judgments.

---

> ### Author Response · Authors · 2025-11-22
> **Response to Reviewer 8HQQ (Part 2)**
>
> 2. Multiple NLI backbones and confidence intervals
>
>     In our paper, ES is computed using the base NLI model `sileod/deberta-v3-base-tasksource-nli`. We further strengthen this analysis by:
>
>     1. adding a larger NLI model `sileod/deberta-v3-large-tasksource-nli`, and
>     2. introducing an LLM-as-judge variant(gpt-4o-mini) that directly decides entailment.
>
>     Because ES is the average of a binary entailment indicator over a finite evaluation set, its estimate is subject to sampling variability even when the generator and NLI backbone are deterministic. We therefore report **95% confidence intervals** using non-parametric bootstrap: for each model, we resample the $N$ evaluation examples with replacement $B = 1000$ replicates, recompute ES, and take the 2.5 and 97.5 percentile values. This captures uncertainty due to the finite benchmark size without assuming any parametric form.
>
>     The results are summarized in **Table R3**:
>
>     Table R3. ES of the original model and 12 unlearning methods on Llama-3 8B Instruct, evaluated using a base NLI model, a large NLI model, and an LLM-as-judge.
>
>     |   Method   | ES – base NLI |    95% CI     | ES – large NLI |    95% CI     | LLM as judge |
>     |:----------:|:-------------:|:-------------:|:--------------:|:-------------:|:-------------:|
>     |  Original  |     0.70      | [0.68, 0.73]  |      0.72      | [0.69, 0.74]  |     0.62      |
>     |  GradDiff  |     0.03      | [0.02, 0.04]  |      0.02      | [0.02, 0.03]  |     0.01      |
>     |    NPO     |     0.02      | [0.01, 0.03]  |      0.02      | [0.01, 0.03]  |     0.01      |
>     |   SimNPO   |     0.00      | [0.00, 0.00]  |      0.00      | [0.00, 0.00]  |     0.00      |
>     |  NPO+SAM   |     0.00      | [0.00, 0.00]  |      0.00      | [0.00, 0.00]  |     0.00      |
>     |  NPO+IRM   |     0.00      | [0.00, 0.00]  |      0.00      | [0.00, 0.00]  |     0.00      |
>     |    RMU     |     0.16      | [0.14, 0.18]  |      0.17      | [0.15, 0.19]  |     0.12      |
>     |     RR     |     0.18      | [0.16, 0.20]  |      0.20      | [0.18, 0.22]  |     0.15      |
>     |    ELM     |     0.38      | [0.35, 0.40]  |      0.40      | [0.37, 0.43]  |     0.31      |
>     |  RMU+LAT   |     0.21      | [0.19, 0.23]  |      0.21      | [0.19, 0.23]  |     0.17      |
>     |    TAR     |     0.00      | [0.00, 0.00]  |      0.00      | [0.00, 0.00]  |     0.00      |
>     |  IDK + AP  |     0.07      | [0.06, 0.09]  |      0.07      | [0.06, 0.09]  |     0.08      |
>     |    DPO     |     0.24      | [0.22, 0.27]  |      0.26      | [0.24, 0.29]  |     0.25      |
>
>     Across all three entailment backbones (base NLI, large NLI, and LLM-as-judge), the **relative conclusions are stable**. The confidence intervals are tight and do not alter the qualitative ranking of methods.

---

> ### Author Response · Authors · 2025-11-22
> **Response to Reviewer 8HQQ (Part 4)**
>
> **Q3.** Extend bit-widths and include multiple schemes (GPTQ/AWQ/RTN). Plot full UE–UT curves to disentangle genuine robustness from capacity collapse, and define an explicit “unusable” region (e.g., $\text{UT}_\text{Open-QA}$ below a percentile threshold) to prevent misleading gains in UE from capacity loss.
>
> **A3.** Thank you very much for this helpful suggestion. In our experiments, the quantization is implemented using **4-bit RTN**, following the setting in [1]. Table 2 in [1] systematically compares different post-training quantization methods and finds that **GPTQ and AWQ perform similarly to RTN** in terms of downstream performance. Motivated by this observation, we focus on RTN as a representative scheme in our study, rather than duplicating experiments over three nearly equivalent quantizers.
>
> To address your request on extending bit-widths, we additionally report results for **no quantization**, **8-bit quantization**, and **4-bit quantization** for both NPO and RMU. The results are summarized in **Table R4 and R5**:
>
> Table R4. NPO under different quantization bit-widths.
>
> |      Metric       | w/o Attack |  8bit  |  4bit  |
> |:-----------------:|:----------:|:------:|:------:|
> | UE$_{\text{MCQ}}$ |   0.27     |  0.27  |  0.27  |
> | UE$_{\text{Open-QA}}$ |   0.02     |  0.02  |  0.00  |
> | UT$_{\text{MCQ}}$ |   0.50     |  0.50  |  0.50  |
> | UT$_{\text{Open-QA}}$ |   0.03     |  0.03  |  0.00  |
>
> Table R5. RMU under different quantization bit-widths.
>
> |       Metric        | w/o Attack |  8bit  |  4bit  |
> |:-------------------:|:----------:|:------:|:------:|
> | UE$_{\text{MCQ}}$       |   0.27     |  0.27  |  0.27  |
> | UE$_{\text{Open-QA}}$   |   0.02     |  0.02  |  0.00  |
> | UT$_{\text{MCQ}}$       |   0.50     |  0.50  |  0.50  |
> | UT$_{\text{Open-QA}}$   |   0.03     |  0.03  |  0.00  |
>
> We observe that **8-bit quantization has almost no effect** on UE/UT compared to the w/o-attack setting, while **4-bit quantization is clearly more aggressive**, mainly driving the Open-QA metrics toward zero.
>
> Regarding the UE–UT curves of quantization, our paper already **plots the full UE–UT trade-off curves for all methods** (Figure 3). Finally, on defining an explicit “unusable” region: we agree that it is conceptually appealing to mark a region where the model is no longer practically usable. However, this threshold could be  highly **task- and application-dependent** (e.g., different practitioners may have very different tolerance levels for UT on Open-QA vs. MCQ). For this reason, we refrain from imposing a single hard-coded percentile threshold in the main paper and instead present the complete evaluation landscape.
>
> For the jailbreaking attack, we follow the setting in [2] and use Enhanced GCG to generate adversarial prompts. As noted in Section 4.4 of [2], other jailbreak prompt strategies rarely succeed against unlearned models, making them ineffective for assessing the robustness of unlearning methods. For this reason, we adopt Enhanced GCG in our evaluation.
>
> > [1] Zhang, Zhiwei, et al. "Catastrophic failure of llm unlearning via quantization." arXiv preprint arXiv:2410.16454 (2024).
> >
> > [2] Łucki, Jakub, et al. "An adversarial perspective on machine unlearning for ai safety." arXiv preprint arXiv:2409.18025 (2024).
> >
> > [3] Cohen, Jacob. "A coefficient of agreement for nominal scales." Educational and psychological measurement 20.1 (1960): 37-46.

---

> ### Author Response · Authors · 2025-11-26
> **Gentle Follow-Up on Rebuttal**
>
> Dear Reviewer 8HQQ,
>
> A few days have passed since we submitted our responses. We are writing to kindly follow up and check whether you have any additional questions or comments, or if our responses have already addressed your concerns. We would be happy to continue the discussion during the open-review period, and we hope that our detailed clarifications help convey the quality and contributions of our work more compellingly.
>
> Thank you again for your time, consideration, and engagement.
>
> Sincerely,
>
> The Authors

---

> > ### Comment · Reviewer_8HQQ · 2025-11-26
> >
> > Thanks for your response and experiments. Your clarifications on the ES reference policy (now consistently tied to benchmark ground truth), plus the human-agreement analysis, multi-judge validation, and CIs, substantially alleviate my concerns about measurement validity. The added cross-model check (Qwen-2.5-14B) and the 8-/4-bit quantization results also make sense and move in the right direction.
> >
> > That said, one critical gap remains:
> >
> > Aggregation stability of metrics. The paper still lacks a robustness analysis for UE_Avg / UT_Avg (standardization before averaging, alternative MCQ:OpenQA weights, rank stability via Kendall-τ/Top-k, and uncertainty bars). Without this, method/family rankings may be artefacts of aggregation.
> >
> > Bottom line: while the study is comprehensive, it reads closer to a **survey-style synthesis** and the actionable insights for method design remain limited. After re-considering the novelty in Sections 4 and 5 again and again, I’m willing to raise **Contribution** from 2 (fair) to 3 (good).

---

> > > ### Author Response · Authors · 2025-11-26
> > >
> > > We sincerely appreciate that our previous rebuttal has substantially helped in addressing your concerns.
> > >
> > > Regarding the aggregation issue, we are happy to further strengthen our robustness analysis. However, we would like to clarify that the aggregated metrics UE_Avg and UT_Avg in **Figure 1(c)** are primarily used as a visualization to illustrate the overall UE–UT trade-off across methods. Our insights are not derived solely from these aggregated values. In fact, the detailed, pre-aggregation metrics have already been fully presented in **Figures 1(a) and 1(b)**, with Figure 1(c) serving only as a complementary summary plot.
> > >
> > > We also note that several aspects of robustness have already been addressed in both the paper and the rebuttal. In particular, **Table R3** reports confidence intervals for UE measured by ES, and ES is evaluated using multiple NLI models (Base NLI Model / Large NLI Model / GPT-4o-mini), all of which confirm the same qualitative findings.
> > >
> > > In addition, we would like to reiterate that our work is not a survey paper. Rather, it presents several actionable and novel insights that directly inform the unlearning method design, such as:
> > >
> > > 1. The taxonomy unifies and clarifies the major families of unlearning approaches. Importantly, it does more than simply categorize; this structure is essential to our study, as it enables a systematic examination of each category’s practical advantages, limitations, trade-offs, and relationships.
> > > 2. The contribution of establishing Open-QA as an essential evaluation dimension beyond traditional MCQ settings (Sec. 4). This expands the evaluation landscape to more realistic and challenging unlearning scenarios.
> > > 3. New insights from fine-grained weight perturbation analysis, contrasting in-domain versus out-of-domain perturbations, and the joint examination of weight-level and input-level robustness (Sec. 5).
> > >
> > > These contributions go beyond summarization and provide concrete, novel understanding that can guide future unlearning algorithm development.
> > >
> > > We again thank the reviewer for the improved Contribution score from 2 (fair) to 3 (good) and the constructive suggestions. We hope that our additional clarifications further strengthen your assessment of our paper.

---

### Official Review · Reviewer_kNrs · 2025-11-08

**Soundness:** 4
**Presentation:** 4
**Contribution:** 2
**Rating:** 4
**Confidence:** 5

**Summary:**

This work is trying to organize the fragmented literature in llm unlearning by
1. developing a taxonomy of twelve unlearning methods into three methodological families: divergence-driven optimization, representation misalignment, and rejection-based targeted unlearning
2. rethinking unlearning evaluation through open-question answering (Open-QA) metrics that complement standard multiple-choice (MCQ) measures
3. providing a multi-axis robustness study spanning in-domain relearning, out-of-domain fine-tuning, quantization, and jailbreak attacks.

An important contribution of the paper is that MCQ-only evaluation yields a myopic view of forgetting behavior, and that robustness properties differ across algorithmic families.

**Strengths:**

1. I like the framing of methods around the three-family taxonomy. It helps unify results from recent lines of work (NPO, RMU, IDK, TAR, etc.) that previously appeared disjoint.
2. Introducing Open-QA metrics exposes phenomena such as over-forgetting and under-forgetting invisible to MCQ scores. Though the use of ROUGE score based evaluation has been a common trend in evaluating unlearning right from the earliest benchmarks such as TOFU. I would frame it as an analysis of the metric, rather than an introduction of it.
3. The decomposition of robustness into in-domain vs out-of-domain fine-tuning, quantization, and jailbreak is valuable and empirically well-motivated.
4. I liked the Loss landscape discussion and how different unlearning methods influence the smoothness, or locality of the change. This could be echoed more!

**Weaknesses:**

The paper’s contribution overlaps conceptually with Wang et al. (ICLR 2025) (Towards Effective Evaluations and Comparisons for LLM Unlearning Methods) and Dorna et al. (NeurIPS 2025) (OpenUnlearning: Accelerating LLM Unlearning via Unified Benchmarking of Methods and Metrics), both of which explicitly define faithfulness and robustness as desiderata for unlearning evaluation.
Yet the present work neither cites nor situates itself relative to those frameworks. This is problematic, and a careful comparison along those axes is important.

### Faithfulness and Robustness
While the paper provides valuable extension to evaluations beyond MCQ to Open-QA, its claims of improved evaluation quality should be examined through the lens of faithfulness and robustness as formalized in prior works such as Wang et al. (2024) and Dorna et al. (2025).
In particular, it remains unclear whether the proposed Open-QA metrics exhibit higher faithfulness to true unlearning outcomes (e.g., measured via calibration or counterfactual red-team tests) and whether they are robust under metric-perturbation protocols. Evaluating this framework within the framework of Wang et. al., or the OpenUnlearning benchmark suite would provide quantitative evidence for its added merit.

### Past contributions
Many of the contributions in this paper, such as use of jailbreak prompts, or quantization have been done in Wang et al (not cited), and the results around extractive and open ended metrics being more robust and faithful also exists in these works. I am having a hard time situating the contributions of this paper, which in isolation is a fantastic collection and systemization of knowledge, to be clear.

**Questions:**

-

---

> ### Author Response · Authors · 2025-11-22
> **Response to Reviewer kNrs (Part 1)**
>
> **Q1.** Response on related work.
>
> **A1.** We thank the reviewer for highlighting the recent related works by Wang et al.[1] and Dorna et al. [2], and we apologize for the oversight in not citing them. Both papers make valuable progress in unifying evaluation practices for LLM unlearning and formalizing desiderata such as faithfulness and robustness. Wang et al. propose the UWC framework and identify Extraction Strength as a core metric, with experiments primarily conducted on the TOFU benchmark to enable fair comparison and calibration of different unlearning methods. Their focus is on how to evaluate and compare methods under a unified metric and calibration scheme. Although they also employ techniques such as jailbreak prompts, these are used to assess the robustness of the evaluation metrics themselves, rather than the robustness of the unlearning methods. Dorna et al.’s OpenUnlearning provides a standardized platform that integrates multiple unlearning methods and evaluation metrics, and further assesses metric faithfulness and robustness through stress tests, with a primary focus on the TOFU benchmark.
>
> **However, our work differs from the prior work in our focus and contributions.**
>
> 1. Our goal is not only to evaluate but to provide a full-stack, methodology-guided analysis of LLM unlearning, from categorizing method families, to rethinking UE/UT evaluation, to analyzing robustness through the lens of these methodological principles and their relationships.
> 2. Novel contribution in a principled taxonomy of unlearning methods (Sec. 3). Neither Wang et al. nor Dorna et al. propose a principled categorization of stateful unlearning methods. Our Section 3 provides a taxonomy of twelve representative methods, grouped into: divergence-driven optimization, representation misalignment, and rejection-based targeted unlearning. This taxonomy enables and guides us to reason about why different methods behave differently under MCQ vs Open-QA, and why their robustness profiles diverge across model-level and input-level perturbations. For instance, rejection-based methods, which receive minimal attention in [1, 2], exhibit unusually strong Open-QA performance despite low MCQ performance (Lines 334-349).
> 3. Novel contribution on Open-QA as an important lens beyond MCQ (Sec. 4). While Wang et al. and Dorna et al. discuss evaluation faithfulness, neither performs a systematic MCQ vs Open-QA analysis grounded in method categories. Our work shows that: MCQ-based UE often overstates unlearning success, Open-QA reveals over-forgetting or utility loss not detectable by MCQ, and method families behave fundamentally differently under these two lenses.
> 4. Novel contribution on finer-level weight perturbations (in-domain vs. out-of-domain) and joint analysis of weight-level and input-level robustness (Sec. 5). Existing works evaluate robustness primarily as a metric property. Yet, we instead analyze method-level robustness by jointly examining: in-domain relearning, out-of-domain fine-tuning,  quantization, and input-level jailbreak attacks. And we connect these robustness dimensions back to the method families introduced in Sec. 3. This yields new insights, such as: divergence-driven optimization is typically more resilient to in-domain relearning; representation misalignment is more resistant to out-of-domain fine-tuning; jailbreak robustness aligns more closely with in-domain relearning than with out-of-domain fine-tuning. These cross-method, cross-metric insights are not explored in Wang et al. or Dorna et al.
> 5. Benchmarking choice on WMDP. Finally, we deliberately focus on WMDP and use it as our primary experimental benchmark because it does not require additional fine-tuning on the forget corpus prior to unlearning, thereby aligning more closely with real-world deployment where base models are used as-is. In contrast, benchmarks like TOFU and MUSE require fine-tuning on domain-specific corpora (e.g., synthetic authors or Harry Potter text) before unlearning, which can degrade general capabilities and introduce overfitting and domain shift, complicating fair unlearning assessment.
>
> As shown in **Table R1**, we report the utility of the original LLaMA-2-7B model and of models fine-tuned on TOFU and MUSE Books (which will be used for unlearning), following the settings of the original papers. We evaluate utility on GSM8K and IFEval, and observe that the TOFU- and MUSE-finetuned models perform substantially worse than the original LLaMA-2-7B. We were concerned that using such already-degraded models as the starting point for unlearning could create a false impression of “poor” utility after unlearning and an artificially inflated sense of “good” unlearning performance when presenting our insights.
>
> Table R1. GSM8K and IFEval of original and TOFU- and MUSE-finetuned LLaMA-2-7B
> | Method  | GSM8k | IFEval |
> |:-------:|:-----:|:------:|
> | Original| 0.23  |  0.44  |
> |  TOFU   | 0.10  |  0.13  |
> |  MUSE   | 0.08  |  0.24  |

---

> ### Author Response · Authors · 2025-11-22
> **Response to Reviewer kNrs (Part 2)**
>
> **Q2.** While the paper provides valuable extension to evaluations beyond MCQ to Open-QA, its claims of improved evaluation quality should be examined through the lens of faithfulness and robustness as formalized in prior works such as Wang et al. (2024) and Dorna et al. (2025). In particular, it remains unclear whether the proposed Open-QA metrics exhibit higher faithfulness to true unlearning outcomes (e.g., measured via calibration or counterfactual red-team tests) and whether they are robust under metric-perturbation protocols. Evaluating this framework within the framework of Wang et. al., or the OpenUnlearning benchmark suite would provide quantitative evidence for its added merit.
>
> **A2.** Thank you for the valuable suggestion. We respectfully argue that the frameworks of Wang et al. [1] and Dorna et al. [2] are not directly applicable to our setting.
>
> Wang et al. select metrics under the key assumption that
>
> > “Accordingly, although values may change, the relative rankings (i.e., the orders of superiority across unlearned models) remains the same without skewing.” (Sec. 3)
>
> This ranking-preservation assumption is central to their metric selection. However, this assumption does not hold in our setting for two reasons: (1) Broader and more diverse method coverage and (2) finer-grained robustness dimensions where rankings naturally diverge. First, our study includes a wider spectrum of unlearning methods, including those with explicit robustness-enforcing designs, such as TAR [4] and NPO+SAM [5]. These methods are specifically engineered to behave differently under model perturbations, and as a result, their relative rankings can vary substantially across different evaluation modalities (MCQ vs. Open-QA) and across robustness conditions. Second, our work analyzes robustness along finer-level axes, e.g., in-domain relearning vs. out-of-domain fine-tuning, which expose fundamentally different vulnerabilities (Sec. 5). Consequently, judging metric faithfulness by whether it preserves a single global ranking is neither realistic nor well-aligned with the nature of these robustness phenomena. For these reasons, the “ranking-preservation” criterion used by Wang et al. is less meaningful for us. Instead, we focus on how Open-QA metrics reveal qualitatively different failure modes, e.g., over-forgetting and generation collapse, that MCQ metrics cannot capture (Fig. 1), and how these behaviors systematically differ across methodological families (Sec. 3). This complements, rather than conflicts with, the faithfulness/robustness discussions in [1,2].
>
> The work by Dorna et al. [2] is also not directly aligned with our setting. Their benchmark pipeline assumes that the base model is first trained or fine-tuned on the forget corpus in order to embed the target knowledge prior to unlearning. This design is appropriate for TOFU and MUSE, where the forget set is deliberately injected through pre-training or fine-tuning. In contrast, WMDP does not require any such pre-unlearning setup: the initial/reference mode is the original chat model, which already possesses extensive knowledge related to WMDP domains (biology, chemistry, cybersecurity) without any additional fine-tuning (as described in our response to Q1).
>
> Following the reviewer’s suggestion, we further validated the behavior and reliability of our Open-QA ES metric through additional analyses.
>
> 1. Agreement with human annotation on WMDP
>
>     We first compare Open-QA ES against human judgments on WMDP using Llama-3 8B Instruct. For each question, we obtain:
>     - a binary ES decision (“unlearned” vs. “not unlearned”)
>     - a binary human label.
>
>     This yields the following confusion matrix:
>     |                      | ES: Unlearn | ES: Not Unlearn |
>     |:--------------------:|:-----------:|:---------------:|
>     | **Human: Unlearn**   |    871      |       15        |
>     | **Human: Not Unlearn** |    20       |      367        |
>
>     The overall accuracy is $\text{Acc} = \frac{871 + 367}{871 + 15 + 20 + 367} = \frac{1238}{1273} \approx 0.9725,$ showing that ES agrees with human annotation **97.25%** of the time, i.e., ES is an extremely high-quality automatic judge. We further compute Cohen’s κ to account for agreement beyond chance. Cohen’s kappa is a statistic that quantifies how much two raters (or judges) agree when classifying items into categories, beyond what would be expected by chance [5].
>
>     Marginal counts:
>     - Human Unlearn = 886
>     - Human Not Unlearn = 387
>     - ES Unlearn = 891
>     - ES Not Unlearn = 382
>
>     The expected agreement under random matching is $p_e = \frac{886 \cdot 891 + 387 \cdot 382}{1273^2} \approx 0.578.$ The observed agreement is $p_o = 0.9725.$ Thus, $\kappa = \frac{p_o - p_e}{1 - p_e} = \frac{0.9725 - 0.578}{1 - 0.578} \approx 0.935.$ A κ of 0.935 falls into the **“almost perfect agreement”** regime in the Landis & Koch scale, indicating gold-standard level consistency between ES and human judgments.

---

> ### Author Response · Authors · 2025-11-22
> **Response to Reviewer kNrs (Part 3)**
>
> 2. Multiple NLI backbones and confidence intervals
>
>     In our paper, ES is computed using the base NLI model `sileod/deberta-v3-base-tasksource-nli`. We further strengthen this analysis by:
>
>     1. adding a larger NLI model `sileod/deberta-v3-large-tasksource-nli`, and
>     2. introducing an LLM-as-judge variant(gpt-4o-mini) that directly decides entailment.
>
>     Because ES is the average of a binary entailment indicator over a finite evaluation set, its estimate is subject to sampling variability even when the generator and NLI backbone are deterministic. We therefore report **95% confidence intervals** using non-parametric bootstrap: for each model, we resample the $N$ evaluation examples with replacement $B = 1000$ replicates, recompute ES, and take the 2.5 and 97.5 percentile values. This captures uncertainty due to the finite benchmark size without assuming any parametric form.
>
>     The results are summarized in **Table R2**:
>
>     Table R2. ES of the original model and 12 unlearning methods on Llama-3 8B Instruct, evaluated using a base NLI model, a large NLI model, and an LLM-as-judge.
>
>     |   Method   | ES – base NLI |    95% CI     | ES – large NLI |    95% CI     | LLM as judge |
>     |:----------:|:-------------:|:-------------:|:--------------:|:-------------:|:-------------:|
>     |  Original  |     0.70      | [0.68, 0.73]  |      0.72      | [0.69, 0.74]  |     0.62      |
>     |  GradDiff  |     0.03      | [0.02, 0.04]  |      0.02      | [0.02, 0.03]  |     0.01      |
>     |    NPO     |     0.02      | [0.01, 0.03]  |      0.02      | [0.01, 0.03]  |     0.01      |
>     |   SimNPO   |     0.00      | [0.00, 0.00]  |      0.00      | [0.00, 0.00]  |     0.00      |
>     |  NPO+SAM   |     0.00      | [0.00, 0.00]  |      0.00      | [0.00, 0.00]  |     0.00      |
>     |  NPO+IRM   |     0.00      | [0.00, 0.00]  |      0.00      | [0.00, 0.00]  |     0.00      |
>     |    RMU     |     0.16      | [0.14, 0.18]  |      0.17      | [0.15, 0.19]  |     0.12      |
>     |     RR     |     0.18      | [0.16, 0.20]  |      0.20      | [0.18, 0.22]  |     0.15      |
>     |    ELM     |     0.38      | [0.35, 0.40]  |      0.40      | [0.37, 0.43]  |     0.31      |
>     |  RMU+LAT   |     0.21      | [0.19, 0.23]  |      0.21      | [0.19, 0.23]  |     0.17      |
>     |    TAR     |     0.00      | [0.00, 0.00]  |      0.00      | [0.00, 0.00]  |     0.00      |
>     |  IDK + AP  |     0.07      | [0.06, 0.09]  |      0.07      | [0.06, 0.09]  |     0.08      |
>     |    DPO     |     0.24      | [0.22, 0.27]  |      0.26      | [0.24, 0.29]  |     0.25      |
>
>     Across all three entailment backbones (base NLI, large NLI, and LLM-as-judge), the **relative conclusions are stable**. The confidence intervals are tight and do not alter the qualitative ranking of methods.

---

> ### Author Response · Authors · 2025-11-22
> **Response to Reviewer kNrs (Part 4)**
>
> **Q3.** Many of the contributions in this paper, such as use of jailbreak prompts, or quantization have been done in Wang et al (not cited), and the results around extractive and open ended metrics being more robust and faithful also exists in these works. I am having a hard time situating the contributions of this paper, which in isolation is a fantastic collection and systemization of knowledge, to be clear.
>
> **A3.** Thank you for the comment. As clarified in Responses #1–2, the perceived overlap with Wang et al. stems from focusing on a few shared “tools” (e.g., jailbreak prompts, quantization) rather than on the “purpose” and “insights” of our study. Our work is not a rebenchmarking exercise: its contributions lie in the methodology-guided and previously overlooked aspects of unlearning (Secs. 3–5). These core insights were not addressed in prior work and were likely overlooked in the reviewer’s assessment. We apologize for any lack of clarity in the original submission, and we hope that our responses help the reviewer better appreciate the methodological perspective and new insights our work contributes.
>
> > [1] Wang, Qizhou, et al. "Towards effective evaluations and comparisons for llm unlearning methods." arXiv preprint arXiv:2406.09179 (2024).
> >
> > [2] Dorna, Vineeth, et al. "OpenUnlearning: Accelerating LLM Unlearning via Unified Benchmarking of Methods and Metrics." arXiv preprint arXiv:2506.12618 (2025).
> >
> > [3] Tamirisa, Rishub, et al. "Tamper-resistant safeguards for open-weight llms." arXiv preprint arXiv:2408.00761 (2024).
> >
> > [4] Fan, Chongyu, et al. "Towards llm unlearning resilient to relearning attacks: A sharpness-aware minimization perspective and beyond." arXiv preprint arXiv:2502.05374 (2025).
> >
> > [5] Cohen, Jacob. "A coefficient of agreement for nominal scales." Educational and psychological measurement 20.1 (1960): 37-46.

---

> ### Author Response · Authors · 2025-11-26
> **Gentle Follow-Up on Rebuttal**
>
> Dear Reviewer kNrs,
>
> A few days have passed since we submitted our responses. We are writing to kindly follow up and check whether you have any additional questions or comments, or if our responses have already addressed your concerns. We would be happy to continue the discussion during the open-review period, and we hope that our detailed clarifications help convey the quality and contributions of our work more compellingly.
>
> Thank you again for your time, consideration, and engagement.
>
> Sincerely,
>
> The Authors

---

### Meta-Review · Area_Chair_uYhJ · 2026-01-06

**Summary:**

This paper presents a comprehensive analysis of LLM unlearning, including a taxonomy of 12 stateful methods, an evaluation beyond MCQ via Open-QA metrics, and a broad robustness study on WMDP. The work is technically careful and the rebuttal addresses several important concerns regarding metric validity, including clarification of the entailment reference, human agreement analysis, multi-judge validation, and additional model-scale experiments.

However, despite these strengths, the reviewer consensus is that the contribution is closer in nature to a survey or benchmarking study rather than a main-track research paper. The core results primarily systematize, compare, and reinterpret existing methods and evaluation practices, with limited new algorithmic or conceptual advances beyond what has recently appeared in evaluation frameworks, unified benchmarks, and survey-style works on LLM unlearning. While the proposed taxonomy is helpful for organizing the literature, it does not on its own provide sufficiently novel insights or methodological advances at the level expected for the research track.
In addition, several central conclusions are tightly coupled to the chosen benchmark and a small number of model configurations. Although the authors justify this choice, the lack of direct comparison with widely used unlearning benchmarks further reinforces the impression that the work functions primarily as an analytical benchmark study rather than a broadly generalizable research contribution.

Overall, the paper is well executed and likely valuable to the community as a reference or evaluation resource, but it is not well aligned with the novelty expectations of the main track. We encourage the authors to consider submitting this work to a dedicated benchmark, dataset, or survey-oriented track, where its strengths in organization, evaluation, and empirical analysis would be better matched.

**Reviewer Concerns:**

The rebuttal effectively addressed several technical concerns raised by the reviewers. The validity of the Open-QA entailment score were largely resolved. Concerns about evaluation scope were also mitigated to some extent by adding results on an additional model.

However, several concerns remain outstanding. The primary issue of novelty was not fully resolved.

**Reviewer Scores:**

Reviewer VkyJ: This reviewer increased their score substantially after the rebuttal.

---

### Decision · Program_Chairs · 2026-01-26

Reject